# A single cell genomics atlas of the *Drosophila* larval eye reveals distinct photoreceptor developmental timelines

Komal Kumar Bollepogu Raja[1,4], Kelvin Yeung[1,4], Yoon-Kyung Shim[1], Yumei Li [2,3], Rui Chen [2,3] & Graeme Mardon [1,2] ✉

The *Drosophila* eye is a powerful model system to study the dynamics of cell differentiation, cell state transitions, cell maturation, and pattern formation. However, a high-resolution single cell genomics resource that accurately profiles all major cell types of the larval eye disc and their spatiotemporal relationships is lacking. Here, we report transcriptomic and chromatin accessibility data for all known cell types in the developing eye. Photoreceptors appear as strands of cells that represent their dynamic developmental timelines. As photoreceptor subtypes mature, they appear to assume a common transcriptomic profile that is dominated by genes involved in axon function. We identify cell type maturation genes, enhancers, and potential regulators, as well as genes with distinct R3 or R4 photoreceptor specific expression. Finally, we observe that the chromatin accessibility between cones and photoreceptors is distinct. These single cell genomics atlases will greatly enhance the power of the *Drosophila* eye as a model system.

Biological tissues with complex mixtures of cellular identities, as well as tissues that show rapidly changing temporal patterns of gene expression often require investigations at single cell resolution for deep mechanistic understanding. Recent technological advances have enabled profiling of transcriptomics, epigenomics, and chromatin configuration from complex tissues at single cell resolution and have transformed our understanding of biological processes. Indeed, single cell molecular atlases of tissues and organs from many species, including numerous model organisms[1–5], have been recently published and are now an essential resource for understanding conserved mechanisms underlying cell fate specification, differentiation, and cell state transitions, as well as the discovery of previously unknown cell types. One of the most commonly used model organisms is *Drosophila*, which has been extensively employed for more than a century to study genetics, development, neuroscience, aging, disease, and many other processes[6–9]. The *Drosophila* eye has been of particular utility since it is easily assayed as an externally visible organ in living animals and has served as a powerful genetic screening tool for decades. The

spatiotemporal nature of the larval eye disc is a unique feature that allows researchers to study developmental dynamics within a single tissue preparation while obviating much of the need for interpolation. Furthermore, many genes involved in retinal determination, such as *Pax6* (*eyeless* in flies), are highly conserved between humans and flies and are required for eye development in both species[10–12]. Therefore, it is of great importance to establish high-resolution single cell genomic atlases for the *Drosophila* eye.

The adult *Drosophila* compound eye is made of ~750 repeating hexagonal units called ommatidia. Each ommatidium has eight photoreceptors (R cells), four non-neuronal lens-secreting 'cone' cells, and six pigment cells. The eye develops from a neuroepithelial sac called the eye imaginal disc during larval and pupal stages. A wave of differentiation called the morphogenetic furrow (MF) begins at the posterior margin of the early larval eye disc and moves anteriorly, leaving differentiating cells behind it. A new column of ommatidia emerges from the MF every 2 hrs such that each column is developmentally more mature than the one immediately anterior to it. Posterior to the

[1]Department of Pathology and Immunology, Baylor College of Medicine, One Baylor Plaza, Houston, TX 77030, USA. [2]Department of Molecular and Human Genetics, Baylor College of Medicine, One Baylor Plaza, Houston, TX 77030, USA. [3]Human Genome Sequencing Center, Baylor College of Medicine, One Baylor Plaza, Houston, TX 77030, USA. [4]These authors contributed equally: Komal Kumar Bollepogu Raja, Kelvin Yeung. ✉e-mail: gmardon@bcm.edu

MF, the R8 cell differentiates first and is the founder cell for each ommatidium. This is followed by the progressive recruitment and differentiation of two pairs of R cells (R2/5 and R3/4). All remaining undifferentiated cells undergo a single round of division, known as the second mitotic wave (SMW). R1/6, R7, and cone cells differentiate after the SMW, while pigment cells differentiate during pupal development.

As a consequence of this progressive cell recruitment to each ommatidium, posterior columns in the eye disc are more mature than their immediate anterior neighbors (Fig. 1B, B'). Therefore, cells in the larval eye disc are arranged in a developmental space-time continuum. More mature differentiating and progenitor cells are found toward the posterior of the eye disc while less mature uncommitted progenitor

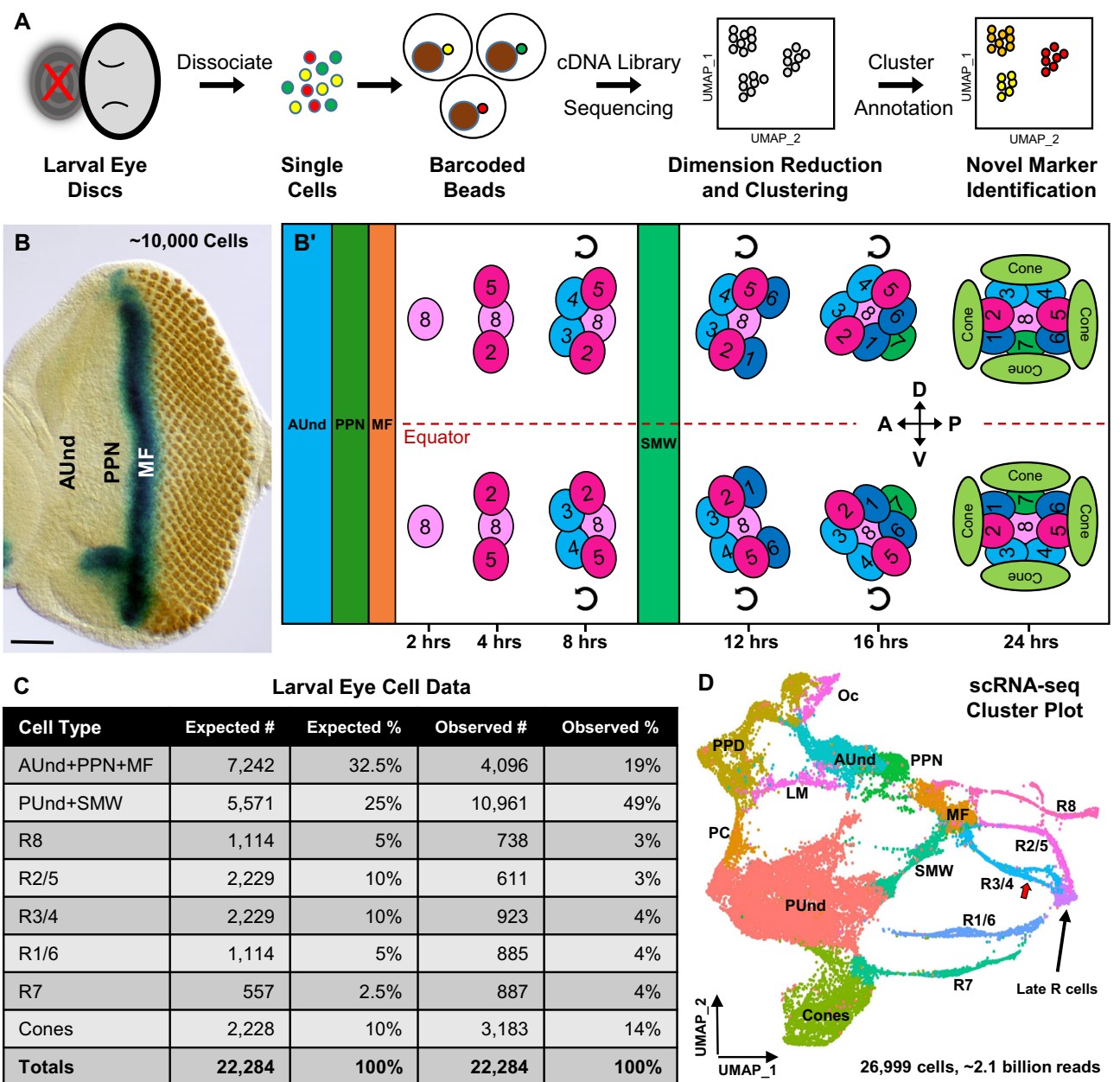

Fig. 1 | Single cell RNA sequencing of the developing late larval *Drosophila* eye disc reveals all expected cell identities. A Schematic of single cell RNA sequencing data generation and analyzes. The red 'X' indicates that the antennal disc was discarded during dissection. B Larval eye disc carrying a *dpp-lacZ* reporter construct were stained with the neuronal marker Elav (rust-colored dots) and β-galactosidase to visualize *dpp* expression (blue). *dpp-lacZ* is expressed in the MF. Scale bar: 50 μm B'. Schematic depicting the arrangement of cell types according to their developmental age in the larval eye disc. Each cluster of cells represents one ommatidium; a total of 12 ommatidia are depicted. Anterior/left to the MF, AUnd and PPN cells are undifferentiated and poised to begin differentiation. Posterior/right to the MF, the R8 photoreceptor differentiates first, followed by R2/5 and R3/4. All undifferentiated cells then undergo one more round of cell division termed the second mitotic wave (SMW). Following the SMW, R1/6, R7 and cone cells are then recruited. The equator is shown as a red dashed line. Dorsal and ventral

ommatidia rotate in opposite directions and exhibit chirality. The direction of rotation is shown as semi-circle arrows. The approximate timing of events is shown below the schematic where t = 0 is when cells first exit the MF. C The expected and observed numbers and percentages of cells for each cell identity are shown. Non-eye disc clusters (PPD, PC, LM, and Oc) were excluded and only major cell type numbers were used for percentage calculations. '#' indicates number and '%' is percentage. D scRNA-seq cluster plot generated from ~27,000 late larval eye disc cells (after filtering to remove low-quality and non-eye disc cells) shows all expected cell identities. Clusters appear in a temporal progression from left to right. The red arrow points to the R3 and R4 'split' in the R3/4 strand. AUnd Anterior Undifferentiated, PPN Preproneural, MF Morphogenetic Furrow, SMW Second Mitotic Wave, PUnd Posterior Undifferentiated, PC Posterior Cuboidal Margin Peripodium, LM Lateral Margin Peripodium, PPD Anterior Peripodial, and Oc Ocelli.

cells are located anteriorly, poised to undergo differentiation (AUnd cells). This progressive, temporal component is a unique aspect of larval eye development compared to most other *Drosophila* tissues. Moreover, ommatidia in the dorsal and ventral halves of the eye disc rotate 90° in opposite directions, resulting in dorsal and ventral ommatidia becoming mirror images of each other (Fig. 1B'). Ommatidial rotation is tightly linked to R3 and R4 differentiation and the interaction between the Frizzled/Dishevelled (Fz/Dsh) and Notch (N) signaling pathways is essential for this process[13,14]. Although several components of these pathways are known, genes that show R3- or R4-specific patterns of expression have not yet been identified to the best of our knowledge, although a single R4-specific enhancer has been reported[13]. Although mechanisms of differentiation and development in the larval eye have been studied at depth, much remains to be deciphered. Therefore, single cell genomics resources for the *Drosophila* larval eye that reflect the repeating, highly ordered, and spatio-temporal properties of this tissue will be of great value.

Although single-cell data from the *Drosophila* larval eye have been previously reported[15–17], the data presented in this report differ substantially. Here, we present single-cell transcriptomic and chromatin accessibility data from late larval eye discs that comprise a deep representation of all known cell types in this tissue. Our data show that all cell clusters appear in a temporal progression resembling the temporal nature of the physical eye disc. R cell clusters appear as distinct strands of cells that are connected to undifferentiated clusters. Furthermore, R cell strands show similar transcriptomes as they mature. We identify dozens of cell type-specific markers, maturation genes, and enhancers, including genes that distinguish the R3 and R4 cells, and also present in vivo validation of many such markers. Finally, we observe that the cone cell cluster shows more specific peaks than R cells. Our high-resolution data provide an invaluable platform for investigating the *Drosophila* eye, as well as greatly aiding research groups that use the eye disc as a model system to study conserved mechanisms of development.

## Results

### Single cell transcriptomics of the developing *Drosophila* eye

We performed single-cell RNA sequencing (scRNA-seq) on two biological replicates of late larval eye discs using the 10x Genomics Chromium platform (Fig. 1A). After removing dead or dying cells, multiplets, and non-retinal cells (e.g., brain and glia), this data set contains 26,999 cells with a sequencing depth of ~2.1 billion reads and 2173 median genes per cell (detailed metrics are provided in Supplementary Fig. 1A). To identify cell clusters, we performed dimension reduction using Uniform Manifold Approximation and Projection (UMAP), a routinely used algorithm in single-cell studies. The UMAP plot shows distinct clusters (cluster plot) corresponding to all expected cell identities in the larval eye disc (Fig. 1D) and each cell type is well represented in both biological replicates (Supplementary Fig. 1L, M). Since a single late larval eye disc consists of ~10,000 cells, the cellular coverage of our data is more than twice the number of cells in the physical eye disc. In addition, as there are only about a dozen different cell identities at this stage, we expect that each cell type is well represented in our dataset and this is indeed the case (Fig. 1C).

### Annotation of cell clusters using marker gene expression

We assigned cell identities in the cluster plot using marker genes known to be expressed in specific cell types at this stage. For instance, we identified the prepreneural (PPN) cluster using *hairy* (*h*), which is expressed in the PPN and negatively regulates the progression of the MF[18] (Fig. 2A, J). The PPN is a subset of the anterior undifferentiated (AUnd) region that is marked by the expression of *Optix*[19]. While *Optix* is expressed in all AUnd cells (including the PPN, Fig. 2B, J), h expression is restricted to the PPN cluster. The

morphogen *decapentaplegic* (*dpp*) is required for the propagation of the MF and its expression marks the MF and lateral margins (LM) of the eye disc[20,21] (Figs. 1B, 2C, J).

We identified three cell populations that correspond to the first R cell subtypes to differentiate (R8, R2/5 and R3/4; Fig. 1B'). These R cell clusters appear as thin strands of cells connected to the MF. One of the strands shows *atonal* (*ato*), *senseless* (*sens*), and *bride of sevenless* (*boss*) (Fig. 2D, J), which are expressed in R8[22–24] and are required for R8 differentiation, development, and function[24,25]. We therefore identify this cluster as R8. We also identified the R2/5 and R3/4 PR clusters using *rough* (*ro*) and *sevenup* (*svp*) as known markers. *ro* is expressed in R2/5 and R3/4[26] (Fig. 2E, J), whereas *sevenup* (*svp*) is expressed in R3/4 and R1/6[27] (Fig. 2F, J). Our data show that *ro* is expressed in two R cell strands and one of these clusters also coexpresses *svp*. We therefore identify the latter cluster as R3/4 while the other cluster that specifically expresses *ro* but not *svp* represents R2/5. The cluster that expresses *svp* without *ro* expression is annotated as R1/6. We also observe that *Bar-H1* (Fig. 2J) and *Bar-H2* (*B-H1* and *B-H2*), known markers of R1/6[28], are expressed in the R1/6 cluster as well as posterior undifferentiated (PUnd) cells.

The transcription factor *prospero* (*pros*) is expressed in R7 cells and in non-neuronal cone cells which secrete the lens[29] (Fig. 2G, J). We observe that *pros* is expressed in two clusters, one of which also expresses *cut* (*ct*), a known cone cell marker[30]. We therefore assigned *pros*-expressing cells that do not express *ct* as R7 and cells that express *ct* and *pros* as cone cells (Fig. 2H, J). After differentiation of R8, R2/5 and R3/4, all remaining undifferentiated cells undergo another round of division known as the second mitotic wave (SMW). R1/6, R7, and cones differentiate from cells following the SMW. A cluster separating the MF and PUnd clusters shows expression of the known cell cycle markers *Claspin* and *Proliferating cell nuclear antigen* (*PCNA*)[15] and we therefore identify this as the SMW cell cluster (Fig. 2J). As expected, our cluster plot shows that the R1/6, R7, and cone cell clusters do not emerge from the MF but appear from PUnd cells that are adjacent to the SMW cluster. In addition, we observe a group of cells far right of the cluster plot into which most R cell subtypes appear to merge. We named this cluster the 'Late R cell' cluster, which is discussed in greater detail in Supplementary Note 1.

We identified PUnd cells using *lozenge* (*lz*), which is expressed in PUnd cells as well as in R1/6/7 and cones[31,32] (Fig. 2I, J). We also identified cell clusters corresponding to the peripodial membrane (PPD), posterior margin cuboidal cells (PC) and ocelli (Oc) using known marker gene expression (Fig. 1D and Supplementary Fig. 1D–K). The expression patterns of many other genes observed in our data are also highly consistent with published studies (Fig. 2A–J and Supplementary Fig. 1B). Taken together, these results show that our scRNA-seq data accurately represents the endogenous mRNA distribution of all genes examined in the larval eye disc.

Remarkably, each cell cluster in our dataset exhibits a temporal component that closely correlates with the developmental progression of cell differentiation in the physical eye disc. The AUnd cluster is next to the PPN and the MF clusters in the expected order. Our data shows R cell subtype clusters as thin strands of cells, as well as a cone cell cluster, which are distinct from the undifferentiated SMW and PUnd cell clusters (Fig. 1D). Overall, the posterior, differentiated part of the eye disc corresponds to the right side of the cluster plot, while anterior, developmentally less mature cells are toward the left (Fig. 1B, D). Moreover, within each differentiated R cell cluster, cells are positioned in a temporal and developmental progression. As expected, trajectory analyzes also show that pseudotime progresses from the less mature AUnd cluster to the distal tips of the PRs (Supplementary Fig. 1N). Taken together, this single cell transcriptomic atlas comprises all expected cell types in the late larval eye disc and all clusters are unambiguously identified and distinct from one another.

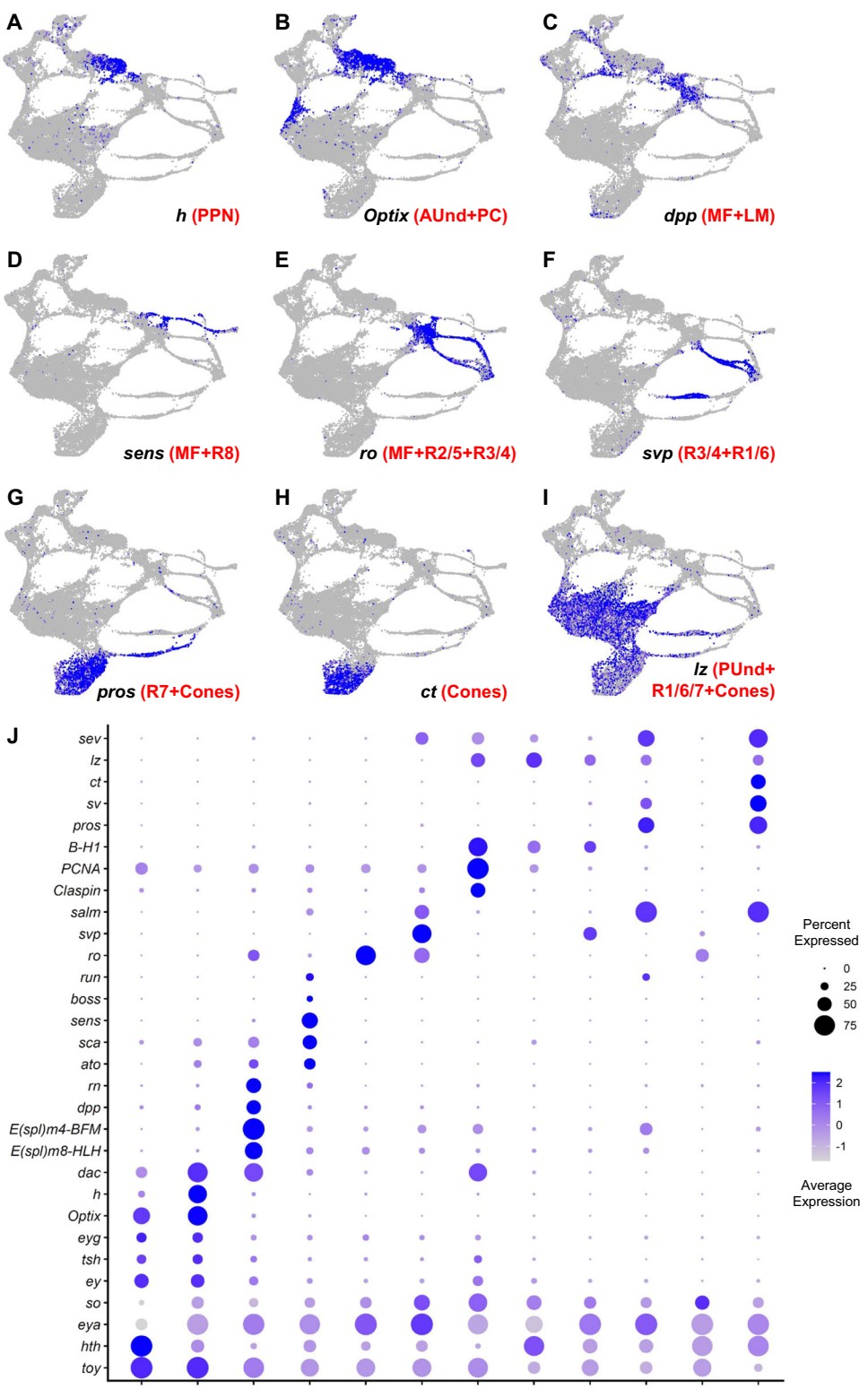

**Fig. 2 | Validation of cluster annotation using known markers. A–I** FeaturePlots showing the expression of marker genes (shown in blue) that were used to identify and annotate clusters. The intensity of blue is proportional to the log-normalized expression levels. **A** *hairy* (*h*) expression is confined to PPN cells. **B** *Optix* expression in AUnd cells. **C** *dpp* expression in the MF and lateral margins. **D** *senseless* (*sens*) expression in the late MF and R8 cluster. **E** *rough* (*ro*) is expressed in the MF, R2/5, and R3/4. **F** *seven up* (*svp*) is expressed in R3/4 and R1/6. **G** *prospero* (*pros*) is expressed in R7 and cone cells. **H** *cut* (*ct*) is expressed in cones. **I** *lozenge* (*lz*) is expressed in PUnd cells and R1/6/7. **J** Dot plot showing the expression of known marker genes. The intensity of blue denotes the average expression level of each gene in each cluster. The size of the circle is proportional to the percentage of cells in each cluster that express the gene. Known marker genes show specific and expected patterns of expression, supporting the assignment of clusters as shown in Fig. 1D.

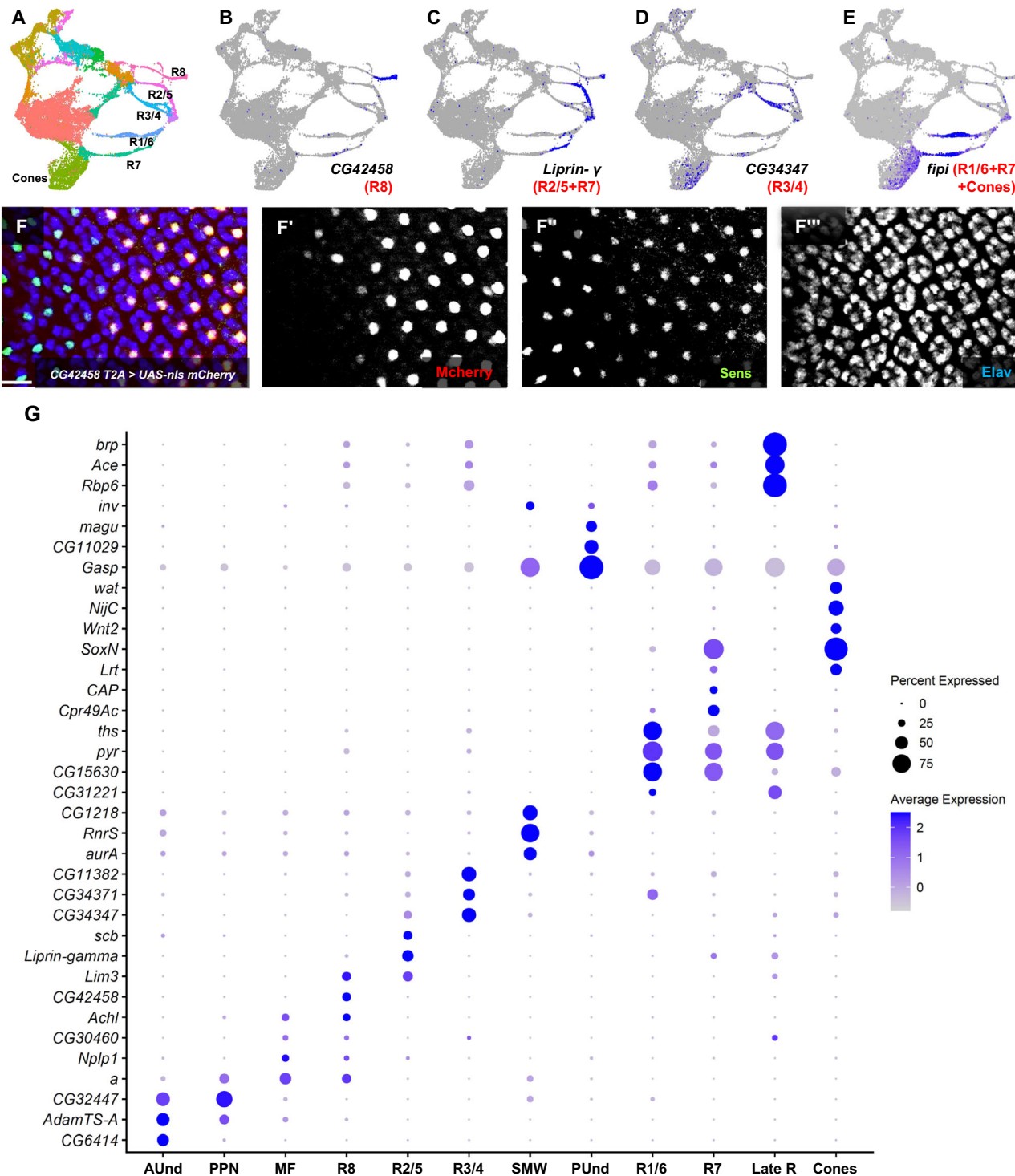

**Fig. 3 | Identification and in vivo validation of markers. A** UMAP cluster plot of the late larval eye disc. **B–E** FeaturePlots showing the expression of cell type-specific genes. **B** *CG42458* FeaturePlot showing expression in R8. **C** *liprin-γ* mRNA is detected in R2/5 and R7. **D** *CG34347* is predominantly expressed in the R3/4 strand. Some expression is also detected in R2/5 and cones. **E** *fipi* expression is observed in R1/6, R7 and a substantial fraction of cone cells. **F–F'''** Staining of eye discs from *CG42458-T2A-Gal4 > UAS-nls-mCherry* larvae showing mCherry expression in Sens-positive cells. Note that *CG42458* is not expressed in early R8 cells (the left side of Panel F'). **G** DotPlot of markers that are highly specific for each cell type in the late larval eye disc. Scale bar: 20 μm.

## Identification of cell type-specific markers

We performed differential gene expression analyzes on all cell clusters and identified cell type-specific markers for each (Supplementary Data 1). Differential gene expression analyzes of the R8 strand reveal 1306 marker genes (Supplementary Data 1). One example is *CG42458*

(Fig. 3B, G); several others are shown in Supplementary Fig. 2A–C. To test if *CG42458* is indeed specifically expressed in R8 cells in vivo, we used a *CG42458-Trojan-Gal4* (*T2A-Gal4*) transgene to drive a nuclear-localized mCherry (*UAS-mCherry-nls*) reporter. *T2A-Gal4* lines carry insertions in genes such that *Gal4* expression is under the control of

endogenous promoters and is driven in a pattern that most often recapitulates the expression of the gene in which the transgene is inserted[33,34]. We costained *CG42458-T2A-Gal4 > UAS-mCherry-nls* larval eye discs with Sens antibody, a known R8 marker (Fig. 3F–F''') and observe that mCherry is present in only one nucleus per ommatidium and colocalizes with Sens. Although mCherry expression starts few columns later than *sens* expression, these data show that *CG42458* is a R8-specific marker, and that the scRNA-seq data accurately represents the endogenous expression pattern of *CG42458*. Similarly, *asense* (*ase*), *CG42313*, and *sidestep* (*side*) show expression in the same cluster as *CG42458* and are likely R8 markers. To our knowledge, none of these four genes have been previously reported to be expressed in R8 cells. We also identified many markers for all other cell types and validated several in vivo (Fig. 3G, Supplementary Figs. 2–4 and Supplementary Data 1).

## Markers that distinguish the R3 and R4 Photoreceptors

Our scRNA data show that the R3/4 cluster emerges from the MF cluster as a single strand but then splits into two smaller strands (Fig. 1D, red arrow). We hypothesized that the R3/4 strand may be splitting into distinct R3 and R4 subtype clusters. Since the two substrands are apparent in both biological replicates, it seemed possible that the split is not an artifact of dimension reduction. We subclustered the R3/4 strand and generated a cluster plot that shows three major clusters (Fig. 4A). These clusters resemble the R3/4 strand shown in Fig. 1D with a single cluster splitting into two subclusters. Differential gene expression analyzes identified several genes that are specifically expressed in only one of the two split subclusters. The genes *prickled* (*pk*), *svp* and *salm* are expressed in both R3 and R4[27,35,36] a few columns posterior to the MF (Figs. 2F, J and 4B, H). We termed the cluster expressing these genes as 'Early R3/4.' Notably, one of the split subclusters shows higher *E(spl)* gene expression (Supplementary Fig. 5B–E). Since Notch signaling is known to be higher in R4 than R3, we hypothesize this subcluster to be R4, while the other could represent R3 cells (Fig. 4A). Differential gene expression analyzes identified several R3- or R4-specific markers (Fig. 4H), including *Dpr-interacting protein δ* (*DIP-δ*) (R3) and *CG4341* (R4).

In larval eye discs, R3 and R4 cells are immediately adjacent, with R4 posterior to R3 in the posterior part of the disc (Fig. 1B')[37]. We drove *UAS-mCherry-nls* using *DIP-δ-T2A-Gal4* and costained larval eye discs with mCherry and Svp. If *DIP-δ* is an R3-specific marker, then we would expect mCherry and Svp coexpression in one cell per ommatidium that is anterior to the other Svp-expressing cell and this is precisely what our staining results show (Fig. 4E–E''). These results suggest that *DIP-δ* is an R3-specific marker (Fig. 4E–E'').

We obtained similar results when we stained *CG4341-T2A-Gal4* driven *UAS-mCherry-nls* eye discs (Fig. 4F–F''). However, in contrast to *DIP-δ*, the mCherry and Svp coexpressing cell is posterior to the other Svp expressing cell, suggesting that it is an R4 cell and that *CG4341* is an R4 marker (Fig. 4F–F''). We also drove *UAS-mCherry-nls* in larval eye discs carrying both *DIP-δ-* and *CG4341-T2A-Gal4* transgenes and observe mCherry and Svp in two cells per ommatidium (Fig. 4G–G''). These data provide strong evidence that these genes are indeed R3- and R4-specific markers. Based on their expression patterns, the genes shown in Fig. 4H are likely to be additional R3- or R4-specific markers. Taken together, these data demonstrate that we successfully profiled gene expression at very high resolution for all cell types present in the larval eye disc.

## Photoreceptor subtypes cluster as discrete temporal strands

Our data show that each R cell cluster appears as a thin strand connected to the MF (R8, R2/5, R3/4) or PUnd cells (R1/6, R7) and all connect at the most mature (posterior/right) ends of the strands to a single putative Late R cell cluster (Fig. 1D and Supplementary Fig. 6A). It is possible that this Late R cell cluster is a vestige of dimension

reduction and further work is required to fully characterize the cells in this cluster. Furthermore, although the R7 and R8 strands do not directly contact the Late R cell cluster, their gene expression patterns at their far right ends (i.e., most mature) largely mimic those seen in the Late R cell cluster (Supplementary Fig. 6I, K). To investigate the temporal pattern of R cell development with higher resolution, we subclustered the MF and R cell clusters and generated a cluster plot that shows the five strands corresponding to R8, R2/5, R3/4, R1/6 and R7 as well as the MF and Late R cell clusters (Supplementary Fig. 6A). The R8, R2/5 and R3/4 strands are connected to the MF cluster, while the R1/6 and R7 strand origins are distinct as they are derived from undifferentiated cells following the SMW. We first compared the expression patterns of the R8 markers *ato*, *sens*, and *boss* (Supplementary Fig. 6B–D) with their in vivo expression patterns[22–24]. We observe a clear progression of expression along the R8 strand from left to right as seen in Supplementary Fig. 6B–D. Specifically, *ato*-expressing cells are within and immediately adjacent to the MF cluster, while *boss*-expressing cells are located at the opposite (right) end of the strand; *sens* expression is flanked by *ato* and *boss* along the strand. These patterns strongly correlate with the known in vivo expression profiles of each gene, suggesting that the strands represent developmental trajectories of individual R cells with cells arranged as a function of time. Developmentally older R cells are located at the posterior ends (i.e., the right side in Fig. 1D and Supplementary Fig. 6A) of each strand while younger cells (i.e., early R cells) are anterior (i.e., left), near the MF. Consistent with this interpretation, we find markers that exhibit the same spatiotemporal dimension in other R cell strands as well. For example, *svp* is expressed in R1/6 in the first few columns posterior to the MF and is absent from R1/6 cells located more posteriorly[27]. *B-H1* expression in R1/6 expression starts at about the same developmental time (i.e., distance from the MF) that *svp* expression stops in R1/6[38]. The distribution of *svp* and *B-H1* mRNA (Supplementary Fig. 7B, C) accurately represent these in vivo expression patterns with *B-H1* RNA detected at the posterior/right side of the R1/6 strand while *svp* RNA is in the anterior/left portion of the strand, with only minimal overlap between the two (black arrows, Supplementary Fig. 7B, C). We also performed trajectory analyzes and observed a clear trend of early to late pseudotime along each R cell strand (Supplementary Fig. 6E). These data collectively show that the R cell strands are two-dimensional developmental trajectories of individual R cells with cells computationally ordered from early to late along each strand.

Since each R cell strand appears to be a linear developmental temporal continuum, PCA on each R cell cluster may unravel the developmental dynamics that drive the maturation of each R cell. The genes with top loadings/weights in the first principal component (PC1) will be those that vary the most in the cluster and are most likely to be acting in the regulatory networks that underlie the maturation of that cell type. We performed PCA on the R8 subcluster and extracted the genes with the highest loadings in PC1 (Supplementary Data 2 and 3). These genes show high variation in the R8 cluster and may be involved in the maturation of R8 cells as they age. Since R cells in the posterior portion of the eye disc are actively sending out their axons and preparing to form synapses, we expect genes related to these processes to be detected in PC1. Indeed, we observe *Down syndrome cell adhesion molecule 2* (*Dscam2*), *golden goal* (*gogo*), *round about 3* (*robo3*), and *Dscam3* in the top 30 PC1 genes; all are known players in axogenesis and axon guidance[39–41]. We also observe several genes whose function in R8 cells is unknown (e.g., *CG42458*, *CG17839* etc.). Moreover, there is a considerable overlap between the top genes PC1 and DE genes of R8, suggesting that genes most enriched in a particular subtype are themselves the most variable in that subtype. We also performed PCA on all other cell clusters and the top genes in PC1 for each are shown in Supplementary Data 2.

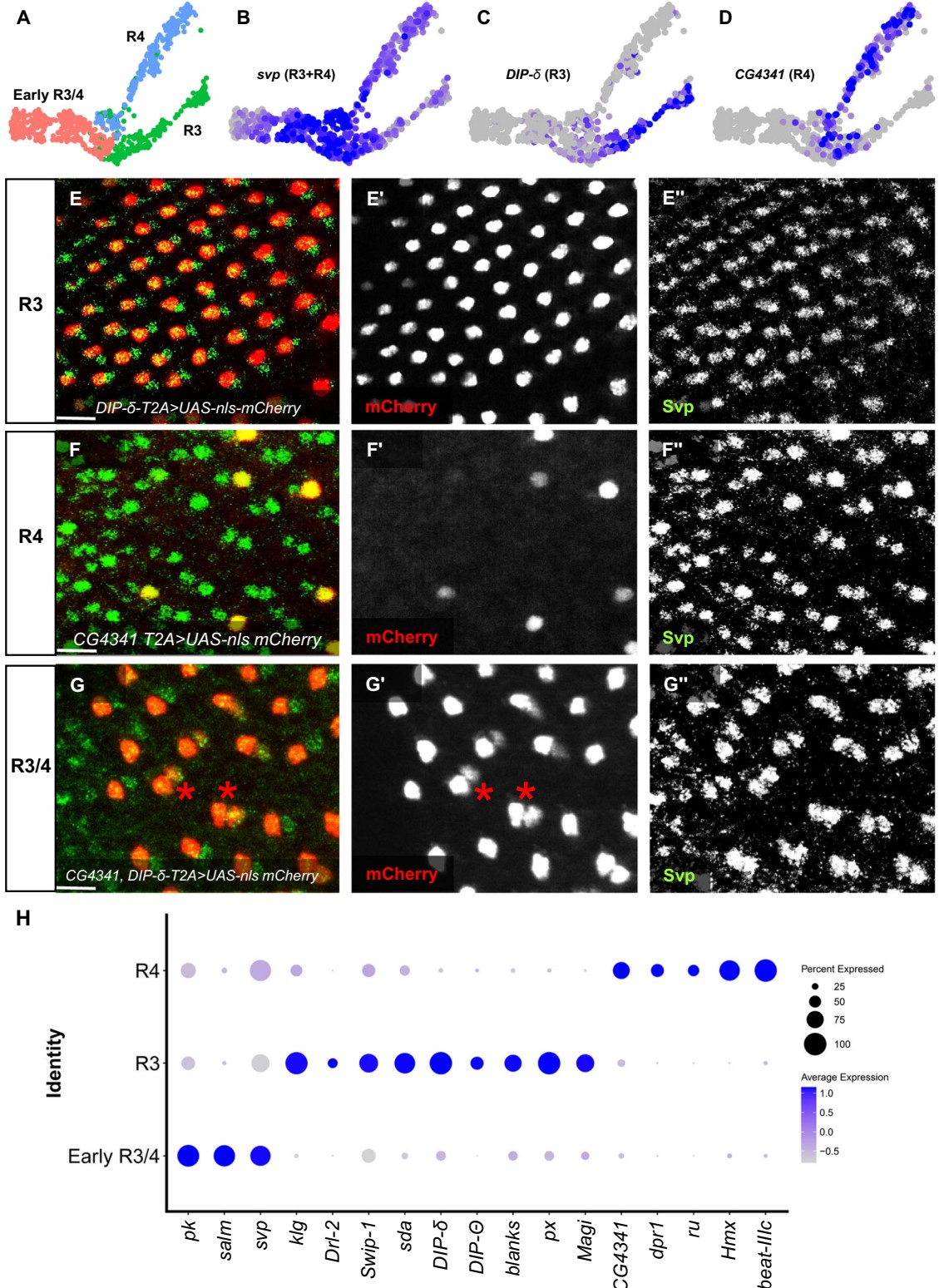

**Fig. 4 | Markers that distinguish the R3 and R4 photoreceptors. A** UMAP cluster plot showing a subset of the R3/4 photoreceptor strand. The R3/4 strand splits into two distinct subclusters. **B** FeaturePlot of *svp* showing expression in both the R3 and R4 strands after the split. **C** *DIP-δ* FeaturePlot showing R3-specific expression. **D** *CG4341* expression is highly specific to the R4 cluster. **E–E″** Larval eye disc image of *DIP-δ-T2A-Gal4 > UAS-nls mCherry* stained with mCherry (red) and Svp (green). Only the anterior (left) of the two Svp-positive cells in each ommatidium costains with mCherry. **F–F″** *CG4341-T2A Gal4 > UAS-nls-mCherry* eye discs stained with mCherry (red) and Svp (green). mCherry is also detected in one cell per ommatidium (although not in all ommatidia), but this costains with the posterior (right) of the two Svp-positive cells in an ommatidium. **G–G″** Larval eye disc carrying both *CG4341* and *DIP-δ-T2A-Gal4* cassettes costained with mCherry (red) and Svp (green). A pair of cells show mCherry expression (asterisk in **G** and **G′**) which also costain with Svp. **H** DotPlot showing markers expressed early in the R3/4 strand as well as R3- and R4-specific markers. All scale bars: 20 μm.

## Single nuclear chromatin accessibility of the larval eye

To complement our scRNA-seq dataset, we performed single nuclear assay for transposase accessible chromatin using sequencing (snATAC-seq) of late larval eye discs using the 10x Chromium Single Cell ATAC Reagent kit (Fig. 5A). We generated sequence data from three biological replicates and the raw data from all three were combined and analyzed, yielding about 29,000 cells. We then used Signac[42] to perform QC, normalization, and dimension reduction steps. After removing non-retinal and peripodial cells, 20,035 cells remained, which is twice the number of cells in a single eye disc. A cluster plot was generated from these cells that shows all the expected cell identities present in the late larval eye disc (Fig. 5B).

Remarkably, the arrangement of clusters in the snATAC-seq cluster plot (Fig. 5B) is highly similar to the clusters seen in the scRNA-seq cluster plot (Fig. 1D). Like scRNA-seq, the snATAC-seq R cell clusters appear as strands of cells, closely resembling the physical eye disc. Overall, the two-dimensional arrangement of all clusters in the snATAC-seq data is nearly identical to that observed from scRNA-seq, including differentiating R cells, AUnd and PUnd cells, as well as the MF, SMW, PPN, and cone cell clusters. R cells appear as thin strands of cells and the R1-R6 strands connect to a common putative Late R cell cluster. The identity of cells in the Late R cell cluster is currently unclear and requires further validation and characterization. Taken together, these snATAC-seq data represent all of the expected cell types with a good representation of each (Fig. 5C).

## snATAC-seq cluster identification and validation

Annotation and analysis of snATAC-seq data presents several challenges compared to scRNA-seq data and the limitations of the current snATAC technology make capturing the entire accessibility profile from an individual cell difficult[43]. snATAC-seq data are relatively sparse and may not reveal cellular variability at many individual regulatory elements, making cluster annotation challenging. We therefore used several approaches to classify and validate cell identities in our snATAC-seq cluster plots. First, we performed integrative analyzes to classify snATAC-seq clusters based on cluster information from scRNA-seq data derived from eye discs at the same developmental stage (Fig. 1D). Using this strategy, cell identity labels were transferred from scRNA-seq to the snATAC-seq cluster map. This method has been used to map cell identities in snATAC-seq clusters from scRNA-seq data in *Drosophila*, humans, and mice[16,44,45]. In addition to transferring labels, gene expression values from scRNA-seq can also be transferred and imputed on snATAC-seq clusters such that marker gene expression patterns can be visualized on snATAC-seq cluster plots. Based on known gene expression patterns, we were able to validate predicted labels on the snATAC-seq clusters by coembedding scRNA-seq and snATAC-seq datasets to visualize both on the same cluster plot[46] (Supplementary Fig. 9).

Second, to test the validity of our snATAC-seq data, we used previously published cell type-specific enhancers to see if our data shows accessible chromatin corresponding to these enhancers. The *dachshund* (*dac*) gene is expressed in the MF, PPN and SMW (Fig. 2J and Fig. 5D). A 3′ enhancer (named '3EE') and a 5′ enhancer (named '5EE') recapitulate the endogenous *dac* pattern of expression in eye discs[47]. The *dac* snATAC-seq genomic track shows an intronic peak and a 3′ peak, which correspond precisely with the 5EE and 3EE enhancers (red bars, Fig. 5E and Supplementary Fig. 10A, A′). Further, the 3′ peak is predominantly accessible in the AUnd, MF, and SMW clusters, reflecting the endogenous pattern of *dac* expression. Similarly, our snATAC-seq dataset shows accessible peaks that correspond to known cell type-specific eye enhancers for *ato, sens, lozenge, shaven* (*sv*), and *pros*[48–52] (Fig. 6A, B, B′ and Supplementary Fig. 10). We also used the JASPAR database[53] to find overrepresented DNA motifs in snATAC-seq peaks. As one example, our motif analyzes show an overrepresentation of the *shaven* (*sv*) motif in both the R7 and cone

cell clusters (Supplementary Fig. 11A-D). *sv* is an R7- and cone-specific marker which regulates neural and cone cell fate decisions in the eye disc[48]. Taken together, these observations suggest that our snATAC-seq data is of high quality, corresponds well with published data, and that many of the cell type assignments have been validated in vivo.

To test if our snATAC-seq data predicts functional enhancer elements, we selected several cell type-specific peaks and used the corresponding DNA to drive reporter gene expression in vivo. The *CAP* gene scRNA-seq and snATAC-seq data show expression specifically in the R7 strand (Fig. 5G, H). Differential accessibility tests of the R7 cell cluster identify a peak in the fifth intron of the *CAP* gene as one of the top peaks. Moreover, the *CAP* locus genomic track reveals a peak that is specific and accessible primarily in the R7 cluster (Fig. 5F). To test if the DNA encompassing this peak contains an enhancer that is sufficient to drive reporter expression specifically in R7, we made reporter constructs with *destabilized GFP* (*dGFP*)[54] driven by this peak DNA and costained transgenic larval eye discs with GFP, Run and Elav antibodies. We observe dGFP expression in a single cell per ommatidium that begins a few columns posterior to Elav expression and costains with the apical Run-positive R7 cell (Fig. 5I–I″). (58). These results show that the fifth intron snATAC-seq peak of *CAP* uncovers an R7-specific enhancer. Similarly, *2mit* mRNA is confined specifically to the Late R cell cluster (Supplementary Fig. 11F, G) and differential accessibility region analyzes of the snATAC-seq Late R cell cluster reveal a peak in the third intron of the *2mit* gene. The CoveragePlot of *2mit* shows that this region is more accessible in the Late R cell cluster compared to other cell identities (Supplementary Fig. 11E). We generated transgenic flies carrying *2mit* snATAC-seq peak DNA upstream of *dGFP* and stained larval eye discs with GFP and Elav. As expected, the peak-region DNA carries a Late R cell cluster-specific enhancer. We observe dGFP expression only in the most posterior columns in late larval eye discs, and dGFP colocalizes with Elav-positive cells (Supplementary Fig. 11H–H″). Taken together, these data show that our snATAC-seq data can accurately predict in vivo enhancer activity and can be used to identify enhancers with temporal and cell-specific properties. Furthermore, these data also validate the cell identity assignments made via label transfer from scRNA-seq to snATAC-seq datasets.

We next performed trajectory analyzes of the snATAC-seq dataset by choosing AUnd cells as the root cells. Although the pseudotime graphs of snATAC-seq (Supplementary Fig. 11I) and scRNA-seq (Fig. 1E) are very similar, one striking difference is apparent: R cells that differentiate prior to the SMW (R8, R2/5, and R3/4) in the snATAC-seq data show a much earlier profile compared to those in the scRNA-seq data. This may reflect the generally more permissive chromatin observed for the R cells compared to cells that differentiate following the SMW (R1/6, R7, and cones). These results suggest that the chromatin accessibility dynamics of R1/6, R7, and cones may be different from other R cell subtypes.

## Chromatin accessibility of R cells and non-neuronal cones

To explore chromatin accessibility across snATAC-seq cell clusters, we performed differential accessibility tests between cell clusters using default metrics to generate lists of differentially accessible marker peaks for each cell cluster in our snATAC-seq data set (Supplementary Data 4). We observe that the number of differentially accessible peaks is higher in R cell clusters than undifferentiated cells with the AUnd, MF, and PUnd cell clusters which show 2–4 fold fewer differentially accessible peaks than R cells. We also investigated the chromatin profile of R cells and cone cells to determine if there are accessibility differences between neuronal and non-neuronal differentiating cells. We analyzed the top 100 differentially accessible peaks in R cell clusters and found that only 0-15% of peaks are subtype-specific (Supplementary Figs. 12 and 13); instead, most of the top 100 peaks are present in most or all R cells. We also examined accessibility profiles near several known R cell markers (Fig. 2A–I) and found a similarly

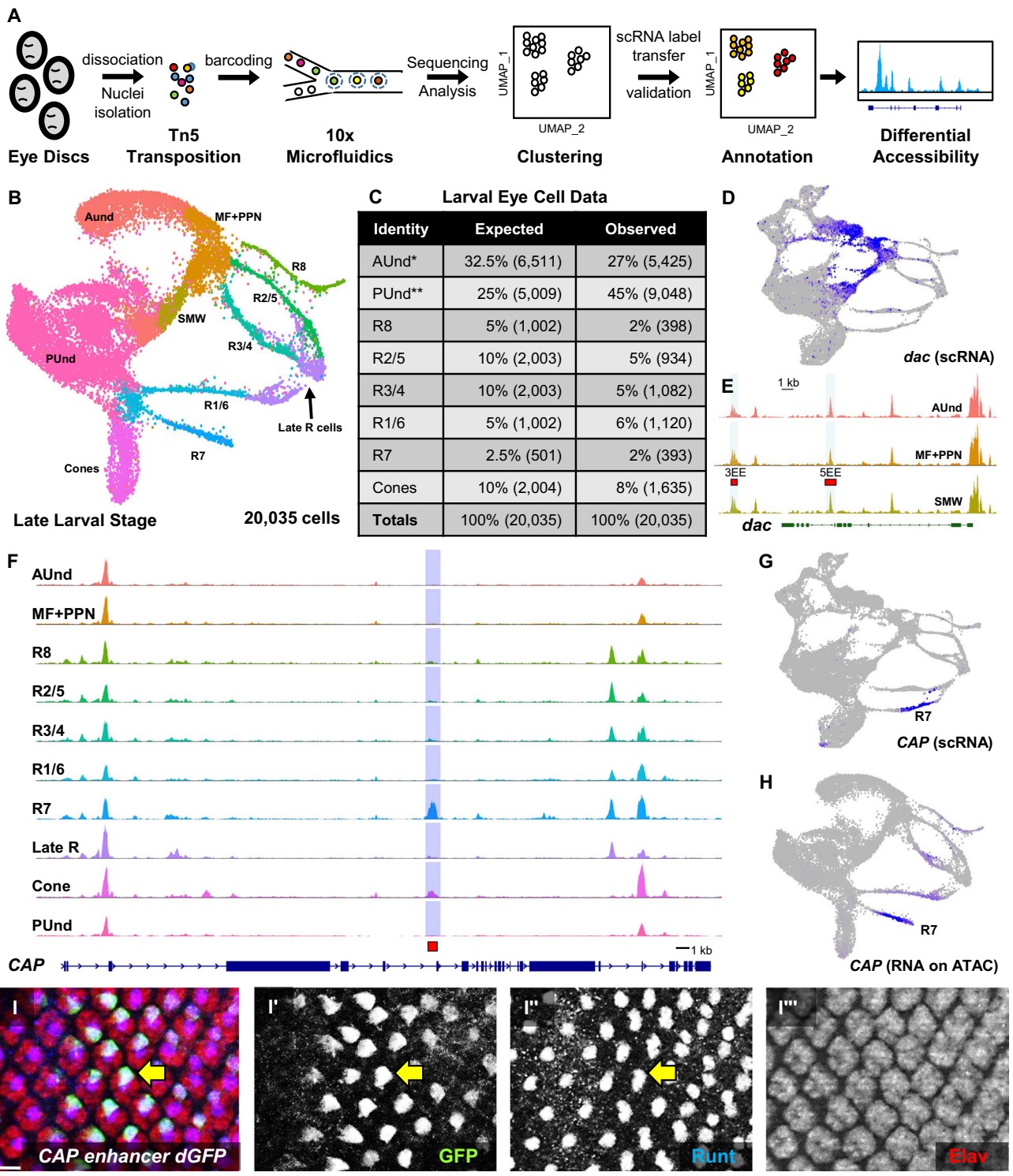

**Fig. 5 | snATAC sequencing of late larval eye discs recapitulates the cellular identities observed in scRNA-seq. A** Schematic of snATAC-seq data generation and downstream analyzes. **B** UMAP cluster plot generated from snATAC sequencing of late larval eye discs shows all expected cell identities. Clusters of cells appear in a temporal progression similar to the UMAP plot from scRNA-seq. **C** The expected and observed cell numbers and percentages for each cell type are shown. Peripodial cells were excluded and percentages were calculated for major cell types in the eye disc.\* = PPN and MF cells are included in AUnd. \*\*= SMW is included in PUnd. **D** scRNA-seq FeaturePlot showing *dac* mRNA expression in the AUnd, PPN, MF, and SMW cell clusters. **E** snATAC-seq CoveragePlot of the *dac* locus for the AUnd, MF + PPN, and SMW clusters. The 3EE and 5EE *dac* enhancers (red bars)

overlap snATAC-seq peaks in these clusters where *dac* is expressed. **F** CoveragePlot of the *CAP* gene reveals a strong snATAC-seq peak highly specific for the R7 and, to a lesser degree, cone cell clusters. The red bar indicates the DNA fragment used to make the enhancer-reporter construct shown in **I–I‴**. **G** FeaturePlot of *CAP* mRNA showing R7-specific expression. **H** FeaturePlot of *CAP* mRNA imputed on the snATAC-seq UMAP cluster plot showing mRNA distribution predominantly in R7. **I–I‴** Costaining of a larval eye disc from an animal carrying the *CAP* R7-specific peak DNA driving a destabilized GFP reporter with GFP (green), Run (blue), and Elav (red) antibodies. One cell in each ommatidium costains with GFP, Run, and Elav (yellow arrow, **I–I‴**). Scale bar: 10 μm.

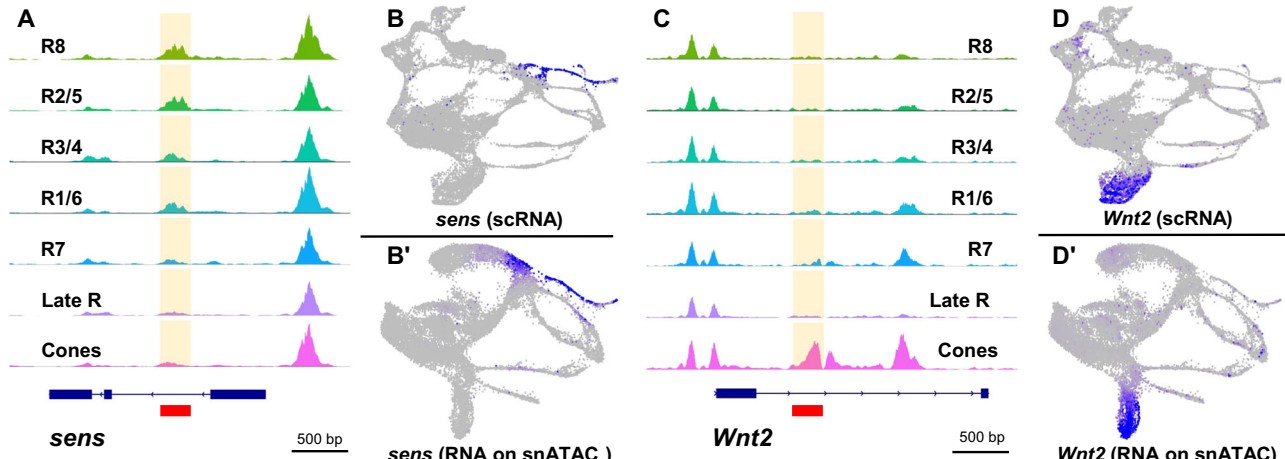

**Fig. 6 | Chromatin accessibility of photoreceptors and cones. A** snATAC-seq CoveragePlot showing the *sens* locus. The solid red bar denotes the *sens* F2 enhancer that is both necessary and sufficient for R8-specific expression in vivo. The snATAC-seq peak is highlighted in yellow. The sens *F2* region is accessible in other R cells in addition to R8. **B** scRNA-seq FeaturePlot of *sens* showing specific expression in the MF and R8. **B'** *sens* mRNA imputed on a snATAC-seq UMAP plot showing distribution of *sens* mRNA specifically in the MF and R8. **C** *Wnt2* CoveragePlot showing a cone-specific peak (solid red bar). The peak is accessible only in the cone cell cluster. **D** scRNA-seq FeaturePlot of *Wnt2* showing mRNA expression is specific to the cone cell cluster. **D'** *Wnt2* RNA overlay on the snATAC-seq cluster plot shows *Wnt2* expression is specific to cones.

permissive chromatin profile across all R cells with little or no specificity for the subtype in which the marker is expressed. For instance, the *sens* F2 enhancer is both necessary and sufficient for R8-specific expression of *sens*. However, the genomic track of *sens* shows that the F2 enhancer region is open in most PRs and not just R8 cells (red box, Fig. 6A). These data suggest that R cell chromatin may be in a largely permissive state with only 3% of R cell genes appearing to show specific snATAC-seq peaks (Supplementary Data 4).

In contrast, among the top 100 snATAC-seq peaks in the cone cell cluster, 70% are predominantly specific to the cone cell identity. For instance, *Wnt2* mRNA is detected only in the cone cell cluster (Fig. 6D, D') and a snATAC-seq peak in the sole intron of *Wnt2* (red bar, Fig. 6C) is one of the top 100 differentially accessible peaks. Moreover, the peak is accessible only in the cone cell cluster. Taken together, these data suggest that there are R cell-specific peaks that are not present in cone cells and vice versa.

### Identifying putative cell type-specific regulators
We used the Single Cell regulatory Network Inference and Clustering (SCENIC) tool to identify important regulators and gene regulatory networks from our scRNA-seq data[55]. SCENIC identifies coexpression modules (termed 'regulons') that comprise sets of genes coexpressed with transcription factors. Our SCENIC results show that R cells cluster separately from other cell types (Fig. 7), suggesting that R cells share similar regulatory networks and cell states that are distinct from other cell types in the eye disc. Furthermore, the observed regulons recapitulate many of the known spatiotemporal gene regulatory network dynamics observed during larval eye disc differentiation. For instance, we observe that the *ato* and *E(spl)m8-HLH* regulons are most active in the MF[25,56] and *svp* in R3/4 and R1/6[27]. We also identified top putative regulators for each cluster by calculating the regulon specificity score (RSS), which is based on the entropy, expression level, and specificity of each regulon in a given cluster (Supplementary Fig. 14A). The regulons *CTCF*, *ato*, *erect wing* (*ewg*), and *RNA binding protein 6* (*Rbp6*) are among the top R8 regulons. Both *ato* and *ewg* are well-known R8 regulators[25,57], but *CTCF* and *Rbp6* are putative R8 regulators identified in this study. Similarly, the regulons *kayak* (*kay*), *longitudinals lacking* (*lola*), *Jun-related antigen* (*Jra*) and *ewg* are among the top R2/5 regulators, whereas *svp*, *ewg* and *lola* are top R3/4 regulators. *kay, lola* and *Jra* have been implicated in eye development[58-60], and *svp* and *lola* are

known to regulate R3/4 fate choice[27,60]. The top putative regulators for all cell clusters are shown in Supplementary Fig. 14A.

Interestingly, the Late R cell cluster on the SCENIC heatmap comprises a remarkably large number of regulons highly upregulated in this cluster compared to all other cell identities (Fig. 7). Our analyzes identified *Rbp6, ewg* and *onecut* as top Late R cell cluster regulators, and expression of *Rbp6* is highly specific to the Late R cell cluster and late R1/6 cells (Supplementary Fig. 14B). Our snATAC-seq motif analyzes using the JASPAR and i-cisTarget databases[53,61] also show Onecut motif enrichment in the Late R cell cluster. *onecut* is highly conserved and required for horizontal cell development in mice[62-64]. In *Drosophila*, we have found that *onecut* null mutants show age-related retinal dysmorphology with loss of R cell rhabdomeres (to be reported elsewhere). Using PANTHER[65], we performed Gene Ontology (GO) term enrichment analyzes with the genes included in each Late R cell cluster regulon and found that GO terms related to axon guidance (GO:0007411) and neuron projection guidance (GO:0097485) are highly enriched. In summary, SCENIC analyzes identified many known regulators of eye development and thereby provided further validation of our scRNA-seq data. In addition, several potentially important regulators were also identified for each cell cluster, which provides new avenues to investigate the regulatory mechanisms underlying eye development.

## Discussion
In this study, we report a comprehensive and high-quality single cell genomics atlas of the developing *Drosophila* late larval eye disc with more than two-fold coverage of the number of cells present in a single disc. Our dissection and dissociation steps were performed in the presence of Actinomycin D (ActD), which inhibits transcription, thereby minimizing stress-related changes to transcription and chromatin configuration[66-68]. Therefore, our scRNA-seq and snATAC-seq data most likely reflect the endogenous gene expression and chromatin configuration of all known cell types in the larval eye disc. Our scRNA-seq data was derived from 26,999 high-quality and viable cells from eye discs with a sequencing depth of 2.1 billion reads, while our snATAC-seq data was generated from 20,595 high-quality nuclei with a sequencing depth of 680 million reads. Our data show distinct cell clusters corresponding to all major cell identities present in the eye disc and each cell type is well represented (Figs. 1D and 5C).

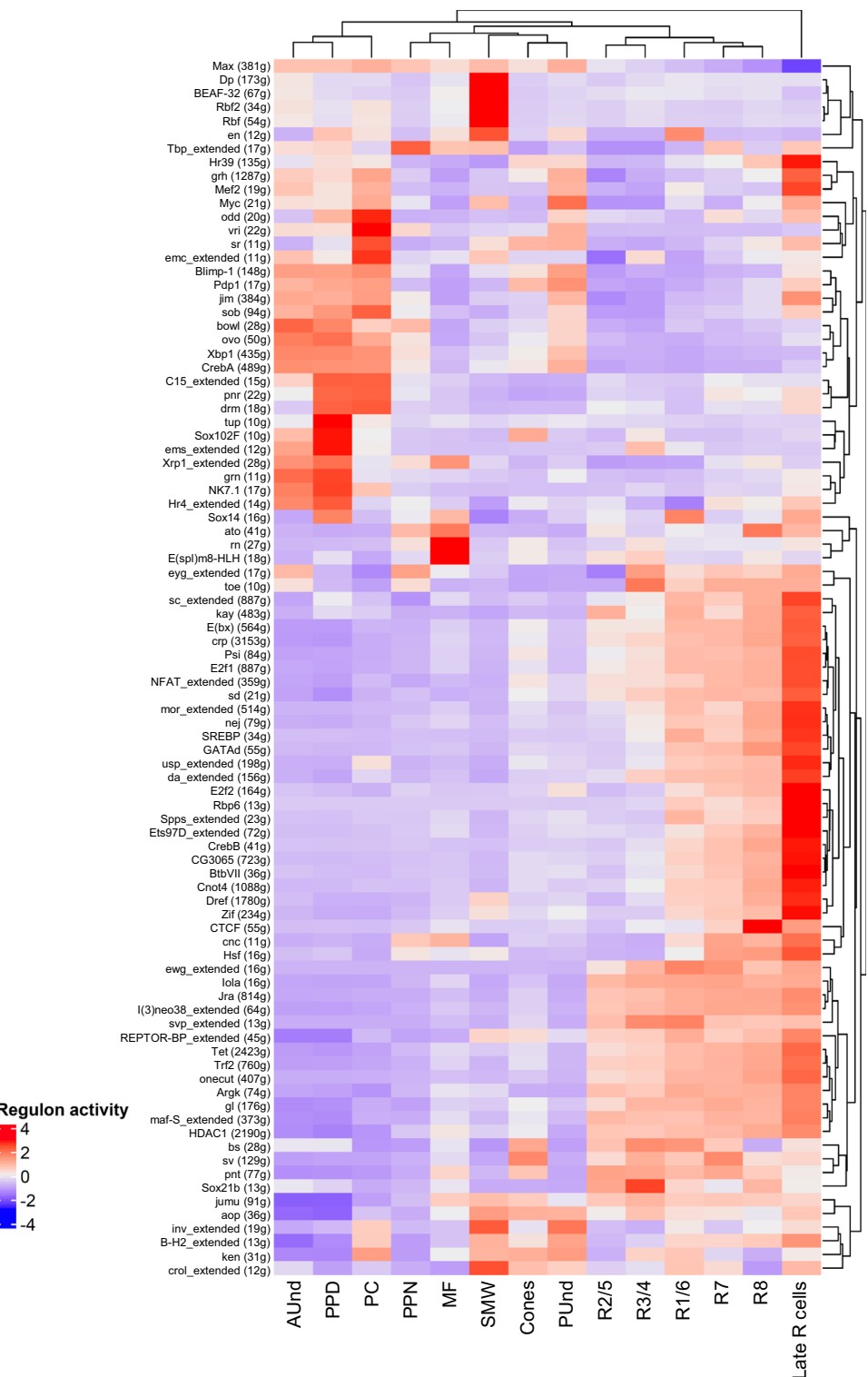

**Fig. 7 | SCENIC analyzes reveals known and putative regulators of each cell identity.** Heatmap showing top regulators of each cell cluster in scRNA-seq. The intensity of red denotes the activity of regulons. The Late R cell cluster is distinct with a large number of upregulated regulons compared to all other cell identities.

Furthermore, both our scRNA-seq and snATAC-seq cluster plots show distinct cell clusters that are ordered sequentially and in a manner consistent with their developmental age. In particular, differentiating R cell clusters are connected to the MF (R8, R2/5, and R3/4) or PUnd (R1/6, R7, and cones) clusters as distinct strands. Moreover, as inferred from known marker gene expression (Supplementary Fig. 6), each R cell strand represents a developmental continuum with cells arranged progressively along the strand with newly differentiating R cells near

the MF and more mature R cells at the far right (posterior) of each strand. Remarkably, cluster plots generated from two completely different types of data (scRNA-seq and snATAC-seq) portray nearly identical arrangements of cell clusters. In addition, analyzes of our data reveal many putative markers, maturation genes, enhancers, and regulators for each cell type, particularly for R cell subtypes and cones, several of which have been validated in vivo. Further, precursor cells of different R cells and cones cluster separately and show distinct

transcriptomic and chromatin accessibility profiles. Investigating the profiles of undifferentiated cell clusters and functional testing of genes in precursor cell clusters may reveal signaling mechanisms and pathways that underlie the specification and differentiation of distinct cell types in the eye disc and are likely to benefit research groups that use the larval eye disc as a model system.

Unexpectedly, we do not observe obvious dorsal-ventral (DV) clustering in our datasets. Specifically, although the gene *mirror* (*mirr*) is known to be highly expressed in the dorsal half of late larval eye discs[69], we do not observe distinct clustering of *mirr*-expressing cells in our data and instead observe *mirr* expression throughout all clusters (Supplementary Fig. 1C). One possibility is that our cellular coverage and sequencing depth may be insufficient to detect transcriptional variation between dorsal and ventral cells. Alternatively, although some genes are clearly expressed either dorsally or ventrally, there are too few such genes at this stage of development to drive the clustering of cells with specific dorsal or ventral identities.

Shortly after R3 and R4 begin to differentiate, dorsal and ventral ommatidia rotate 90° in opposite directions and appear as mirror images of each other in adults. The establishment of this ommatidial orientation requires Fz/Dsh and N signaling between R3 and R4[13,14]. The polarizing signal is initiated in R3 through the transcriptional activation of the N ligand Delta (Dl) by Fz/Dsh. As a result, N is activated in the neighboring R4 cell and therefore has higher N signaling compared to R3. The *Enhancer of split* (*E(spl)*) genes are the downstream effectors of N signaling and a ~500 bp enhancer fragment (named mδ0.5) of *E(spl)mδ-HLH* is the only marker known to differentiate R3 and R4 and drives reporter expression in an R4-specific pattern[13]. Our scRNA-seq cluster plot shows that the R3/4 strand splits into two smaller strands, which we have identified as R3 and R4. As expected, our data shows higher expression of *E(spl)* genes in R4 compared to R3. We also identified several R3- and R4-specific markers and validated *DIP-δ* and *CG4341* as R3- and R4-specific, respectively. Interestingly, these markers are detected in the smaller strands but are not detected in the early R3/4 cluster. We confirmed these expression patterns in vivo for both *DIP-δ* and *CG4341*: reporter gene expression is observed in R3 or R4 in posterior ommatidial columns, but not in early R3/4 cells. This suggests that R3 and R4 are transcriptionally very similar in the first few columns posterior to the MF but become distinct and express different markers further posteriorly. In contrast, our snATAC-seq clustering does not show any R3/4 distinction with the R3/4 strand appearing as a single strand without a split. This suggests that the chromatin profiles of the R3 and R4 strands may be very similar. To the best of our knowledge, this is the only report that identifies genes that distinguish the R3 and R4 photoreceptors and these may provide insights into mechanisms that establish epithelial polarity. We have not yet identified any genes that distinguish individual R cells in the R2/5 and R1/6 pairs in our analyzes. However, since individual members of these R cell pairs are not known to possess distinct functions (unlike R3 and R4), the lack of a transcriptional distinction within these R cell pairs may not be unexpected. Interestingly, we observe *Defective proboscis response* (*Dpr*) gene expression in R3 while their heterophilic binding partners encoded by the *DIP* genes are expressed in R4. DIPs are immunoglobulin-like receptors for Dpr proteins and are often found on interacting neuronal processes. Since signaling between R3 and R4 is well known, it is possible that *Dpr-DIP* gene interactions may be involved in communications between these neighboring cells.

Our scRNA-seq and snATAC-seq clustering results show that R cell clusters R1-6 (and, perhaps to a lesser extent, R7 and R8) appear to converge toward a common putative Late R cell cluster that is far right on the cluster map. This putative Late R cell cluster appears on UMAP in independent datasets that measure very different aspects of the genome (scRNA-seq and snATAC-seq). Currently, the origin of the cells in this cluster is unknown and full characterization of these cells would require further work. However, our data suggest that this cluster consists of Late R cells as they appear at the distal tips of R cell strands and express markers related to axon projection, guidance, and synapse formation. Furthermore, our in vivo data show expression of marker genes in the posterior R cell columns, and similar to the R cell strands, cells in the Late R cell cluster also appear to be ordered in a temporal series that strongly correlates with in vivo gene expression. We have identified many markers and putative regulators that are expressed in the Late R cell cluster, including *Rbp6* and *onecut*, which are well-conserved among animals and play important roles in neural development[70,71].

The R cells exhibit higher transcriptional diversity during early stages of differentiation and then the cell type-specific gene expression appears to become less prominent as they mature and merge in the Late R cell cluster. The phenomenon of neurons exhibiting a common transcriptome during synaptogenesis was reported for *Drosophila* olfactory projection neurons and the optic lobe[72–75]. It has been hypothesized that synapse formation in the brain beyond neuropil targeting may require synchronous wiring of neurons and therefore all such neurons must express similar markers[75]. A similar requirement may pertain to R cells that are projecting their axons to the medulla and lamina of the optic lobe. Early in development, R cell transcriptional distinction may reflect a requirement for precise positioning of each R cell within a single ommatidium. As R cells mature, however, synchronous targeting may be necessary for proper synaptogenesis and therefore may drive highly similar transcriptomes. It is possible that high expression levels of many common genes (e.g., those involved in axogenesis and synapse formation) in Late R cells obscures subtype-specific gene expression such that dimension reduction using UMAP results in a single merged cluster. The absence of cell type-specific genes in the top PC1 genes in this cluster supports this premise. Thus, while subtypes no longer cluster separately and appear to have a common identity, their transcriptional distinctions are likely to persist. Therefore, the Late R cell cluster may consist of multimodal and related R cells. Furthermore, since R1-6 and R7/8 target their axons to different layers of the brain, it may be expected that R7 and R8 clusters do not fully merge with R1-6. Future cell lineage tracing studies can confirm whether the Late R cells show cell type-specific gene expression or assume a common transcriptome.

Our snATAC-seq data shows that there are considerable differences in the chromatin accessibility of R cells and cones. While cones show several snATAC-seq peaks accessible only in the cone cell cluster, the chromatin in R cells is accessible in most or all R cell subtypes, including several peaks that are associated with known marker genes that show very specific subtype expression (e.g., *sens-F2*). Furthermore, cone cell cluster snATAC-seq peaks are often near genes that show cone-specific expression, which is distinct when compared to R cell subtype clusters, which collectively show peaks near genes involved in axon guidance and projection (Supplementary Data 5). Since the chromatin states of R cell subtypes are very similar, cell type differentiation may be largely controlled by distinct transcription factor activity in different R cell subtypes. In contrast, both the chromatin state and localized cone-specific gene expression may be involved in cone differentiation. We also observe that the snATAC-seq peaks associated with R cells are generally not accessible in cone cells, further highlighting the differences in cell states between these two major cell types. We also observe that the frequency of cell type-specific peaks increases in cells that differentiate after the SMW, with cones showing the most specific peaks, followed by R7 and R1/6, suggesting that chromatin is substantially remodeled after cells exit the SMW. Although the mechanisms underlying differences in the chromatin states of R cells and cones is unclear, these results highlight the potential utility of our snATAC-seq data in deciphering cell fate determination and differentiation.

Although single-cell studies on the larval eye disc have been previously reported[15–17], the work presented here has substantially greater depth and resolution (Supplementary Fig. 15). In contrast to these published studies, our data profiles more cells with deep representation of all known cell types in the eye. Our cluster plots show more distinct groups of cells that are virtually arranged as a temporal progression of clusters reflecting normal development: the anterior undifferentiated cell cluster is followed by preproneural cells, the morphogenetic furrow, and all photoreceptor subtypes. Moreover, photoreceptors emerge from undifferentiated clusters as strands of cells that represent the temporal maturation of cells across the disc. Furthermore, all known marker genes examined in our data show the expected patterns of expression with highly cluster-specific distribution. Distinct marker gene expression in specific clusters is not observed in any of the previous studies (Supplementary Fig. 15). In addition, we report identification of many cell type-specific markers that should aid in exploring mechanisms underlying eye development and function. Finally, our data sets identify genes that show distinct R3- or R4-specific expression patterns, which has not been previously reported in the *Drosophila* eye.

In summary, we provide a high-quality and extensive single-cell genomics atlas of the *Drosophila* larval eye that represents all cell types in the eye disc, including the identification of many cell type-specific genes, enhancers, and putative regulators. Moreover, photoreceptor clusters are observed as strands representing developmental continuums that strongly correlate with in vivo gene expression. Intriguingly, these initially distinct R cells strands show a common transcriptomic signature dominated by axon-related genes by late larval stages. Our analyzes of chromatin accessibility show R cell and cone-specific peak profiles. These single-cell resources provide a wealth of genome-wide transcriptomic and chromatin accessibility data that will dramatically aid in investigating mechanisms of cell fate determination, development, and function.

## Methods

### Fly husbandry

All flies used in this report were maintained at 25˚C on cornmeal agar medium. We obtained the following fly stocks from the Bloomington *Drosophila* Stock Center: *UAS-mCherry-nls* (38424), *MiMIC-CG42458* (67472), *CRIMIC-Liprin-gamma* (79357), *MiMIC-CG34347* (76674), *MiMIC-DIP-δ* (90320), *MiMIC-CG4341* (76620), *CRIMIC-chp* (78931), and *CRIMIC-qvr* (86367). The *roFF::GFP* flies were generated by *GenetiVision Corporation* with support from grant R44 GM148146 and based on the protocol described in Manivannan et al. 2019[76].

### Dissociation of late larval eye discs into single cells for scRNA-seq

We collected 25 to 30 eye discs from *Drosophila melanogaster Canton-S* late male larvae (0 hrs after puparium formation) and immediately transferred them to a LoBind 1.5 ml Eppendorf tube containing 700 µl ice-cold Rinaldini solution supplemented with 1.9 µM Actinomycin-D, a known transcription inhibitor. After dissection, 16 µl of collagenase (100 mg/ml; Sigma-Aldrich #C9697) and 2 µl of dispase (1 mg/ml; Sigma-Aldrich #D4818) were added to the tube. The tube was then placed horizontally in a shaker and the eye discs were dissociated for 50 min at 32˚C at 250 rpm. The solution was pipetted every 10 min to disrupt clumps of cells. The cells were then diluted with 1 ml of Rinaldini solution containing 0.05% Bovine Serum Albumin (BSA). The diluted cell suspension was passed through a 35 µm sterile filter and centrifuged at 4 °C at 50 to 100 g to obtain a cell pellet. The pellet was washed once with Rinaldini + 0.05% BSA and subjected to centrifugation. The cell pellet was resuspended in Rinaldini + 0.05% BSA and the viability was assessed using Hoechst-propidium iodide solution. Samples that showed >95% viability were used for scRNA-seq experiments.

### Single cell RNA-seq using 10x Genomics

We used Chromium Next GEM Single Cell 3' Reagent Kit v3.1 from 10x Genomics to generate single cell libraries from single cell suspensions that showed >95% viability and at 1000-1200 cells/µl concentration. Briefly, single cell suspensions were loaded on a 10x Genomics Chromium Controller along with Gel Beads containing barcoded primers and oil emulsion. The Chromium Controller isolates each cell in an oil droplet with a Gel Bead (GEM). The cells are lysed and the mRNA is captured and barcoded within the oil-bead emulsion followed by reverse transcription to synthesize cDNA. The cDNAs from each cell were pooled and a library was generated and sequenced with a NovaSeq 6000 (Illumina). We sequenced two biological replicates to a sequencing depth of ~1 billion reads each. FASTQ files generated from each sequencing run were combined and analyzed using the Cell Ranger v6.0.1 aggr (aggregate) pipeline. The *Drosophila melanogaster* reference genome Release 6 (dm6) was used to make the reference genome using the Cell Ranger 'mkref' pipeline.

### Seurat analyzes

The filtered gene expression matrices from the Cell Ranger output had ~36,000 cells with a sequencing depth of 2.6 billion reads. These cells were used as input to perform downstream analyzes in Seurat v4.03. We first merged the two replicates and removed potential multiplets and lysed cells by retaining cells that showed a total number of genes between 200 and 5000 and cells that showed low mitochondrial gene percentage ( < 30%). The filtered cells were then processed with the Seurat SCTransform algorithm that normalizes and scales the data across all cells. The number of variable features used for SCTransform was 5000 and regression was performed using mitochondrial genes to remove effects of mitochondrial gene expression on clustering. The dimensionality of the data was then reduced using the top 50 dimensions, and the data was clustered using the RunUMAP, FindNeighbors and FindClusters functions in Seurat. Different random seeds (42, 123, 1000, and 2000) and dimensions (10, 20, 30, 40, and 50) were tested for UMAP in the RunUMAP function. Using known markers, we removed cells from antenna (*Distal-less* (*dll*))[77], glia (*reversed polarity* (*repo*))[78] and brain (*found in neurons* (*fne*))[79], and 26,999 cells with a sequencing depth of ~2.1 billion reads were retained from the eye disc proper, Oc, PC and LM, and the PPD. Differential marker gene lists for all cell clusters were generated using the FindAllMarkers function with a log-fold change threshold value of 0.25 and a minimum percentage of cells in which the gene is detected of 25%. For trajectory analyzes, the 'SeuratWrappers' R package was used to convert the scRNA-seq Seurat object to a celldataset object for analyzes in Monocle 3. Cells were ordered in pseudotime by selecting the AUnd cell cluster as the root cells. Similar analyzes were performed for the MF and R cell subclusters with the MF as the root cells. Each cell type was subclustered and PCA was run using Seurat 'subset' and 'RunPCA' functions, respectively. The top 1000 PC1 genes were extracted with loading values for each gene. The FeaturePlot function was used to visualize the expression and distribution of genes. FeaturePlot creates a lattice plot that shows colored single cells on a dimensional reduction cluster plot. DotPlot was used to visualize gene expression across clusters.

### Dissociation of larval eye discs into single nuclei for snATAC-seq

We dissected and collected 30 to 40 late larval eye discs from *Canton-S* males and removed antennal discs prior to transferring them to a LoBind Eppendorf tube containing ice-cold 1x PBS supplemented with 1.9 µM ActD. The 1x PBS was replaced with 100 µl lysis buffer containing digitonin (0.005%) and Nonidet P40 Substitute (Sigma, #74385) and incubated on ice for 5 min. The solution was pipetted every 1 min to lyse the cell membranes and to release intact nuclei into the solution. After 5 min, the lysis reaction was stopped by diluting with Tris-HCl (10 mM, pH 7.4) wash buffer containing 10% BSA. The solution was

then passed through a 10 μm filter to remove cell debris and clumps and subjected to centrifugation at 100 g at 4 °C. The supernatant was discarded and the nuclear pellet was resuspended in Tris-HCl wash buffer and centrifuged one more time at 4 °C to obtain a nuclear pellet. The pellet was then resuspended in 1x Nuclei Buffer (10x Genomics, PN-2000153, PN-2000207) and the integrity of nuclear membranes was ascertained using bright field microscopy. Nuclear samples that showed intact nuclear membranes without blebbing were used for snATAC-seq experiments.

## snATAC-seq using 10x Genomics and downstream analyzes
We used 10x Chromium Next GEM Single Cell ATAC Kit v2 to generate libraries from dissociated nuclei. Briefly, nuclei at a concentration of 3000/μl were used to recover ~10,000 cells for each snATAC-seq experiment. The nuclear suspension was first mixed with the 10x Chromium Transposition Mix and incubated at 37 °C for 30 min for the transposition reaction. The Transposase Mix fragments the DNA and adds adapter sequences to the fragments. The transposed nuclei were then loaded into the 10x Chromium Controller along with barcoded Gel Beads and partitioning oil to generate GEMs. The Gel Beads were then dissolved and the barcoded DNA fragments pooled to generate libraries. We generated libraries from three independent biological replicates and sequenced them separately with a NovaSeq 6000 (Illumina). FASTQ files generated from each sequencing run were pooled and analyzed using the cellranger-atac aggr v2.0 pipeline. The output of the Cell Ranger ATAC pipeline was used as input for Signac v1.8.0 for quality control (QC) and downstream analyzes. We first removed potential multiplets and lysed nuclei by selecting nuclei that showed total peak region fragments between 500 and 30,000. Next, we selected nuclei that showed more than 25% (15% default in Signac) of fragments in peaks, which represents the fraction of all fragments that fall within ATAC-seq peaks. Reads that may be artifactual (blacklist regions) were also removed using the ENCODE dm6 blacklist[80]. The filtered data was then subjected to normalization and dimensional reduction followed by the identification of clusters using the RunUMAP function. We used the top 50 dimensions to perform dimension reduction.

We performed integrative analyzes to classify snATAC-seq clusters based on the cluster information from scRNA-seq data derived from eye discs at the same developmental stage. First, snATAC-seq data was used to quantify the transcriptional activity of each gene in the genome by counting the number of fragments (Gene Activity score) that map to the 2 kb upstream promoter region of each gene. A matrix with these scores and the 2 kb upstream gene coordinates was created. The snATAC-seq gene activity score matrix and the gene expression matrix from our scRNA-seq data were used as input for canonical correlation analysis (CCA), which is a statistical analysis tool used to identify integration anchors between two datasets. Briefly, CCA projects the two datasets into shared dimensional space using gene expression values from scRNA-seq and gene activity scores from snATAC-seq. Cells that share similar biological states or patterns, such as gene expression values, appear together and are referred to as 'mutual nearest neighbors.' Cells that show this pairwise correspondence between scRNA-seq and snATAC-seq are marked as 'anchors.' Using integration anchors, cell identity labels were transferred from scRNA-seq to the snATAC-seq cluster map. In addition, gene expression values from scRNA-seq data were transferred and imputed on snATAC-seq clusters, and marker gene expression patterns were visualized on the snATAC-seq cluster plot. Cell clusters pertaining to the brain, glia, PC, and PPD were removed and only cells from the eye disc proper were retained.

## Immunohistochemistry
Late larval eye discs were dissected and immediately transferred to a 1.5 μl Eppendorf tube containing ice-cold 1x PBS and fixed in 3.7% paraformaldehyde in PBS for 30 min at room temperature. Eye discs were then washed 3 times with PBS + 0.3% Triton X-100 (PBT) and blocked using 5% normal goat serum in PBT. Primary antibody incubations were done overnight at 4 °C. Secondary antibody incubations were performed at room temperature for at least 1 hr. Eye discs were washed and mounted on glass slides for imaging. A Zeiss Apotome Imager microscope was used to generate optically stacked images, which were processed with Zen Blue and Adobe Photoshop software. We used the following antibodies: rat anti-Elav (DHSB-7E8A10, RRID:AB, #52818, 1:500), chicken anti-GFP (Abcam, catalog number: ab13970, RRID:AB, #300798, 1:1000), rabbit anti-mCherry (Thermofischer scientific, catalog number: MA5-47061,RRID:AB, #2889995, 1:2000), guinea pig anti-Runt (gift from Dr. Claude Desplan), guinea pig anti-Sens (a gift from Hugo Bellen, 1:1000) and mouse anti-Svp (DHSB-2D3, RRID:AB, #2618079, 1:500). The following secondary antibodies were used at 1:500 concentration: Cy5 anti-rat (Jackson Immunoresearch, catalog number: 712-175-153, RRID: AB, #2534067), Cy5 anti-guinea pig (Abcam, catalog number: ab102372,RRID:AB, #2340460), Alexa 488 anti-guinea pig (Thermofischer Scientific, catalog number: A-11073, RRID: AB, #2534117), Alexa 488 anti-chicken (Thermofischer Scientific, catalog number: A-11039, RRID:AB, #2762843), Alexa 568 anti-rabbit (Thermofischer Scientific, catalog number: A10042, RRID:AB, #2534017), Alexa 488 anti-mouse (Thermofischer Scientific, catalog number: A-11029, RRID: AB, #2536161) and Alexa 555 anti-rat (Thermofischer Scientific, catalog number: A-21434, RRID: AB, #2535855).

## Transgenic assays to identify functional enhancers
To identify functional enhancers, we selected cell type-specific peaks and designed primers that span the entire 'called' peak sequences. We PCR amplified the DNA corresponding to the peaks and cloned the purified PCR products into *pH-Stinger-dGFP-attB* or *pH-Stinger-mCherry-attB* vectors. We generated transgenic flies using site-specific integration and used the attP2 landing site. Transgenic flies were generated by *GenetiVision Corporation*. Transgenic late larval eye discs were dissected and stained with the antibodies listed in the results as previously described[81]. Eye discs were imaged using a Zeiss Apotome Imager microscope to generate optically stacked images. Zen blue and Adobe Photoshop software were used to process the stacked images.

## GO term analyzes
Genes that were near differentially accessible R cell, Late R cell and cone cluster peaks were used for analysis with Panther. The fold enrichment of the top enriched GO terms for biological processes were used to make bar graphs.

## Statistics and reproducibility
Each scRNA-seq sample was prepared using 25 to 30 late larval eye discs from 15 animals, while 30 to 40 eye discs from at least 20 animals were used for each snATAC-seq sample. More than 20,000 filtered cells were obtained for both scRNA-seq and snATAC-seq experiments. The number of cells in each major cell type cluster is shown in Fig. 1C and Fig. 5C for scRNA-seq and snATAC-seq, respectively. For immunohistochemistry, eye discs from 15 late larvae were dissected and stained. Data with similar results from more than 15 to 20 eye discs only were included.

## Reporting summary
Further information on research design is available in the Nature Portfolio Reporting Summary linked to this article.

## Data availability
The raw and processed data generated in this study have been deposited in Gene Expression Omnibus under Accession number GSE235110.

## Code availability

All R scripts used to generate the data shown in this work were uploaded onto GitHub (https://github.com/komalbollepogu/Drosophila_LarvalEye_SingleCell).

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

## Acknowledgements

We thank Claude Desplan for sharing anti-Runt antibodies and Hugo Bellen for sharing anti-Senseless antibodies. We thank Nick Tran, Shinya Yamamoto, Justin Ma, and Benjamin Frankfort for their insightful comments on our manuscript. This work was funded in part by The Retina Research Foundation. Library prep and sequencing were performed at the Single Cell Genomics Core at Baylor College of Medicine and were partially supported by NIH shared instrument grants (S10OD023469, S10OD025240), P30EY002520, and CPRIT grant RP200504.

## Author contributions

K.K.B.R. and K.Y. performed the larval eye disc dissection and analyzes of scRNA-seq data. The single-cell dissociation protocol and snATAC-seq analyzes were performed by K.K.B.R. Y.L. performed cDNA library construction and sequencing of single-cell cDNA libraries. K.K.B.R. prepared and conducted the immunofluorescence imaging of larval eye discs. Y.S. performed enhancer cloning. The manuscript was prepared by K.K.B.R. and reviewed by K.Y., R.C., and G.M.

## Competing interests

G.M. and R.C. are co-owners of *GenetiVision Corporation*. The remaining authors declare no competing interests.
