## [Peer Review File · Nature Communications]

A Single Cell Genomics Atlas of the Drosophila Larval Eye Reveals Distinct Photoreceptor Developmental TimelinesREVIEWER COMMENTS

Reviewer #1 (Remarks to the Author):

This manuscript is an outstanding example of the power of scRNAseq as both discovery tool and resource. It takes a well studied system, the fly eye, and provides new information about its development. The data is thorough and of first rate quality, greatly superceding the rather mediocre scRNAseq published earlier for the eye. Moreover, they experimentally validate many of their observations using transgenics and imaging. This is first rate work.

As such, this data will be a rich resource for mathematicians, physicists, and data scientists to explore important questions regarding dynamics of cell state transitions, cell maturation, positional information, and pattern formation. Nature Communications is a terrific forum for this paper given the interdisciplinary bent of the journal.

I think the authors actually do a disservice to themselves and the work by not emphasizing the unique property of the eye disc to capture developmental dynamics in one tissue sample without any inference. This is opposite to scRNAseq of embryos or other organs where either single samples are studied or sparse temporal sampling is performed. These studies rely on heavyhanded and questionable methods to mathematically infer the underlying dynamics. These inferences rarely are validated. Whereas here in this manuscript, the dynamics are beautifully laid out with remarkable temporal resolution. The authors should rewrite some of the introduction, results, and discussion to emphasize the singularity of this work.

I also think the authors could easily enhance the impact of their work by some simple linear dimension reduction analysis like PCA. More details below. If they do PCA on subsets of cells as I suggest below, they will definitely be able to publish in this manuscript new insights about gene networks and dynamics of cell maturation and specifiication. If it is fruitful, which I am confident it will be, they could replace some of the more simplistic analysis main figures (Figs 3 and 8) with PCA results. It should only take a couple of weeks to analyze the data by PCA.

Below I list a number of points as I came across them in the manuscript. Some are major points and some minor.

INTRODUCTION

The field has traditionally used the acronym R cells for the 8 photoreceptors instead of PR cells. It makes more sense to keep with the tradition in context of discussing R cell specific transcriptomes. etc. Throughout the manuscript.

The MF generates a new column every 2 hours not 1.5 hours. This had been inferred in 1980s experiments and validated by Gallagher et al in 2022 (elife) in which they observed it in live discs.

It makes more sense to move the sentence about furrow speed to before the one beginning "Posterior to the MF, the R8..." It makes everything following more easy to comprehend

There are R3 or R4 specific genes that have been already described in the literature. E(spl)m-delta for example. Correct this error.

“The R7 PR and four cone cells are added to each ommatidial cluster during late larval stages “ This sentence is misleading. There is a gradient of differentiation where in late L3 larvae, some ommatidia do not have R7 and cone cells. Perhaps it would be more clear to say the R7 and cone cells are recruited to ommatidia that are located more posterior in the eye disc since these ommatidia are more mature than their anterior neighbors.

It states “less mature uncommitted progenitor cells are located anteriorly” This is not strictly correct. The progenitor pool is distributed throughout the eye posterior to the MF - only a minority of progenitors actually differentiate into R cells and cones. There may indeed be stages of progenitor maturation. For example, Yan and Pnt genes are expressed with pulsatile dynamics in progenitors as they age (Pelaez et al 2015 *elife* and Bernasek et al *Biorxiv* <https://www.biorxiv.org/content/10.1101/430744v4>). But we cannot really know if this represents maturation since maturation usually is irreversible. This scRNA seq may actually uncover maturation.

It is repeatedly stated that there are streams or clusters of cells in the abstract and introduction. They are referring to the dimensionally reduced data representations. Calling them streams or clusters prior to showing the data is misleading. The summary paragraph at the end of the introduction might benefit better from less description of what the low dimensional representations look like and more what conceptual advances are made.

Figure 1 refers to AUnd and PPN and SMW but the introduction does not mention these developmental processes. Maybe it would be better to cite Figure 1B and B' in the introduction and also discuss these undescribed processes there. They are described somewhat in the figure legend but it is better for readers to read in main text and the introduction is the appropriate place.

RESULTS

Figure 1 and 2, “streams of cells emerging” implies some sort of dynamics rather than what UMAP is showing which is a low dimensional shape or geometry of the high dimensional data. Strands is a more appropriate descriptor. Strands connected to clusters is also more accurate. Not emergence. Please replace streams with strands throughout.

The analysis of R3 and R4 data is fascinating. In addition to finding expected genes like E(spl), they find enrichment of DPR1 in R4 and DIPs in R3. They do not make note of the literature on these proteins - DIPs are Ig-like receptors for DPRs like DPR1. Often found on interacting neuronal processes, here they are expressed in neighboring cells that are known to signal one another. Is this DPR-DIP as signal between the two as well? This bears discussion.

Fig 1E and text uses a standard “pseudotime” algorithm to infer dynamical features from the static data (snapshot collected). Pseudotime inferences are in general very weak and rarely validated. In this present situation, it is completely unnecessary. Since the eye disc has the unique feature of having cells at progressively older stages of development all contained together, there is no need to use pseudotime. Real developmental time is represented in the data. This is really seen by the interconnected strands of datapoints representing the R cells, SMW cells and LM cells. This is exactly what one predicts from developmental data with embedded dynamical features. For point of reference, please see the Tabula Muris data of mouse hematopoietic cells (<https://www.biorxiv.org/content/10.1101/2021.05.03.442465v1>),

which is also a system with embedded dynamics. Therefore, the pseudotime analysis should be removed and more effort placed on extracting dynamical features of the data (see PCA below).

The representation of the oldest R cells is a cluster (convergence cluster) in the UMAP embedding. The authors then subject the individual classes of R cell data to selective UMAP embedding. While this is one approach to take, they are probably embedding data that might be much more amenable to linear dimension reduction like PCA. UMAP and its ilk are non-linear DR methods that give us a probabilistic and non-quantitative view of high dimensional data. If each R cells' data cloud is more linear in structure, even if high dimensional, then PCA will not only accurately show the cloud's structure, but the distances between datapoints, ie. cells, will quantitatively scale with real measures of difference. This is not the case for UMAP or t-SNE. It would be trivial for the authors to run PCA on the selected R cell data (python, R and Matlab all have PCA packages). In fact if they select each R cell type to analyze by PCA, then the first eigenvector (PC1) will uncover the developmental dynamics of that cell type's maturation program. Genes correlated along that eigenvector having large eigenvalues will be those genes acting in the GRN for maturation of that cell type. The identities and positions of those genes in the GRN will be nicely displayed. The convergence cluster itself could be subject to PCA and genes differentially expressed in the cluster or biased in expression will be identified. It will settle whether the cluster is a unitary "melded" cell type of mature R cells or is a collection of highly related multimodal cells types (I think it is probably the latter).

PCA of any cluster (ie SMW, MF, PPN, etc) can be done to reveal the hidden dynamics of GRNs working in each class of cells. This is easy to do, and is potentially of great value in such a developmental system since it is physically designed for this kind of analysis.

My comments about PCA apply not only to the scRNA data but also scATAC data. This beautiful data shows a similar UMAP embedding representation as scRNA. Classes of cells based on identities could also be subject to PCA as described above for scRNA data. It would then be trivial to correlate genes detected along PC1 of the scRNA data with genes detected along PC1 of the ATAC data. This would help build robust GRNs out of the work. The authors attempt to use SCENIC to start building GRNs but this is a crude tool given the remarkable data at the disposal of the authors.

In this regard, the focus of the manuscript on new "marker gene" identification does disservice to the enormous potential of the eye system to understand the transcriptome dynamics of cell fate transitions and cell state maturation.

Figure 6B - how were cell identities classified since the data is differential genome reads? Did they look for cells where known marker genes had significantly enriched read counts? More information should be given in the text.^[SEP]

DISCUSSION

Line 490: computational representation should be changed to mathematical representation.

Line 509 sentence explains why DV domain cells are not embedded within clusters. Another reason could be that they are embedded but within 3D clusters whereas UMAP embeds in 2D. So a cluster, say dorsal R7s might lie above ventral R7s in 3D space but UMAP squashes both together with its 2D embedding.

Lines 553 -568 is a bit naive. Cells are not “losing their identity” as they mature. They are expressing sets of genes necessary for maturation of all photoreceptor cells - axon guidance, rhabdomere morphogenesis, opsins etc. The few highly distinct TFs expressed early in their maturation are being complemented with many more common genes expressed in neurons and photoreceptors, thus obscuring the differences these cells still have with their distinct TFs expressed early and late. The UMAP embedding is a statistical representation, and correlations are stronger if more genes are expressed in common. Hence there is a single cluster. The same naivete can be said about the olfactory and optic lobe papers.

Reviewer #2 (Remarks to the Author):

This manuscript describes two datasets: a single-cell mRNA sequencing and a single-cell ATAC sequencing dataset of the late third instar larval eye disc. The authors generated these two datasets, performed standard analysis of the two datasets in R that led to the generation of two UMAP plots, which they then used to assign clusters based on known markers. Then, they use this information to identify cell-type specific markers and cell-type specific open chromatin regions, respectively. Finally, they discuss two observations: a) the convergence of all photoreceptors (PRs) in one state and b) the difference in chromatin accessibility between PRs and cone cells.

The generated datasets appear to be of very good quality, although it would have been necessary for the reviewers to have the data and the scripts available to be able to actually review the paper. However, the paper appears to be just a report of the datasets with minimal, if any, new ideas and knowledge.

Moreover, the two observations that are discussed are not well supported and not convincing.

Regarding the convergence of photoreceptors to a common state:

- We know that the photoreceptors are different with one another, especially R7 and R8, which have different molecular identities, they express different rhodopsins (although this expression comes later, the PRs have to keep their identity and not converge with other PRs). The fact that they converge might be an artifact of the analysis. Instead of re-clustering and remaking the UMAP, the authors should, at least, renormalize all expression data, choose new variable genes (i.e. variable genes only within photoreceptors) and see if they still converge. Even then though, does it make any sense that PRs at L3 lose their individual identity? They haven't made any synapse yet.
- What happens later into pupal stages? How does this convergence evolve especially when the PRs have to make synapses or when the R7 and R8 cells acquire a pale or yellow identity?
- The authors should verify the expression of synaptic genes at the level of the protein as well. Isn't it too early to make synapses? Their postsynaptic targets are not mature to make synapses before P20 or P30.
- In any case, what is the “biological importance” of this convergence? What does it mean for the photoreceptor specification? Does it mean that they all have to go through a process that involves axon guidance and neuronal projection? Is this new or unexpected?

Regarding the argument about accessibility differences between PRs and cone cells: this is very unclear and needs to be further developed, tested, and justified. What are the potential

“functionally important differences”? What function do they serve?

Except for the above, I also have two other points that I find concerning:

- First, the authors rely extensively on the use of UMAP to extract information. This is problematic and can lead to false conclusions and confirmation bias by reading the UMAP selectively (<https://www.biorxiv.org/content/10.1101/2021.08.25.457696v1>). For example, the authors don't mention that R7 cells seem to come from cone cells and not the PUnd cluster, which is, of course, biologically incorrect. They also mention that the UMAP is correlated with the physical eye disc, which is in the eye of the beholder. The extensive use of UMAP plots to draw conclusions can be very misleading and has been done extensively in this manuscript.
- It is impossible to review the paper without having access to the data, but more importantly to the scripts, especially since most of the bioinformatic analyses are not described in detail.

Minor points:

- In Figure 8, the authors argue that PRs cluster separately from other cell types. This is actually not true if we believe the dendrogram. In fact, the Convergence cluster (i.e. a PR cluster) is the outgroup of all clusters. What does this mean?
- Mentioning the four classes of DE genes in the text without explaining what they actually are is of little use. The authors should either describe them in the text or not mention them at all.
- Mentioning numbers of DE genes without mentioning any cutoff is also of little use. How were the DE genes chosen? Fold-change, p-value, q-value? It is known that Seurat gives overblown numbers of DE genes.
- The extensive use of R commands, such as FeaturePlot, CoveragePlot, etc, should be eliminated, as this makes the manuscript inaccessible.
- The stage of the sequenced eye disk (late L3) should be mentioned in the Abstract
- In the Abstract, the authors mention that a single-cell genomic atlas will greatly empower the Drosophila eye as a model system. I don't think the eye disk needs such an atlas to be empowered. It has been one of the most commonly used model systems, as the authors also state in the Introduction.

Reviewer #3 (Remarks to the Author):

In this manuscript the authors present a single-cell multi-omics resource detailing the RNA-Seq and ATAC-Seq profiles of the developing drosophila eye. The noteworthy aspects of this work are 1) the number of cells profiled (~27,000) which suggests coverage of all cell types present in the late larval eye, 2) identification of novel cell type markers, and 3) the interesting developmental pattern whereby photoreceptors emerge from transcriptionally distinct trajectories only to converge upon a similar transcriptional identity. The significance of this work is hinted at in the manuscript because of the richness of the drosophila system for understanding the role of developmental genes and gene regulatory networks such as Pax6, but could have reached a more ambitious potential by comparing wildtype patterns of development to mutants affecting eye development including mutants of the potential regulators identified by SCENIC analyses. The methodology is sound and meets expected standards in the field of single cell transcriptomics and epigenomics. There is sufficient detail for the experiments to be reproduced. This reviewer has no major concerns apart from that this work stands mostly as a resource upon which deeper insights may be gleaned in the

future. A minor suggestion for improving the current manuscript would be to condense the part of the manuscript devoted to identifying clusters based on known markers, to leave more room to compare and contrast with the previous single-cell studies performed by this group and others and to emphasize the new found knowledge and understanding of eye development that has been gained by the current study.

Reviewer #4 (Remarks to the Author):

In this work, the authors report a single cell genomics atlas of the *Drosophila* larval eye. Both scRNA-Seq data and scATAC-Seq were generated using the 10X genomics platforms from similar tissues. Overall, I think this will be a useful data resource. The followings are my comments.

1. The variation of biological replicates. The scRNA-Seq dataset was generated from two biological replicates and the scATAC-Seq dataset was generated from three biological replicates. The authors showed through UMAP plots that the major cell types are separated in scRNA-Seq and scATAC-Seq data. In the analysis, the authors focus on the variation of different cell types, which may be confounded by the variation of different biological replicates. It would be great to show UMAP plots, with different colors representing different biological replicates, to see whether the same cell types from different biological replicates are mixed together or not.

2. In this work, many conclusions are drawn based on patterns of UMAP plots: e.g. relationship between different cell types. Because UMAP provides a 2D representation of the original dataset and the result will vary depending on the hyperparameter in UMAP and also the random seed. It will be great if the authors can provide the hyperparameters used in UMAP plots. It would also be great if the authors can test different hyperparameters and random seeds in implementing UMAP and check whether the conclusions in the work is rigorous (especially when the random seed in UMAP changes).

3. In this work, "In addition to transferring labels, gene expression values from scRNA-seq can also be transferred and imputed on snATAC-seq clusters such that marker gene expression patterns can be visualized on snATAC-seq cluster plots." This practice may be problematic for validation: both the cell type labels and gene expression are transferred from scRNA-seq, and it is expected to see agreement between the transferred marker gene expression and the transferred cell type. It seems to me that a more reasonable practice is to use gene activity score of marker genes in scATAC-Seq to validate the cell type labels transferred from scRNA-Seq data.

4. The raw and processed datasets should be accessible to the reviewers. Reviewer tokens in GEO should be given for reviewers to check that the datasets are publicly available.

Reviewer Comments

Reviewer #1 (Remarks to the Author):

This manuscript is an outstanding example of the power of scRNAseq as both discovery tool and resource. It takes a well-studied system, the fly eye, and provides new information about its development. The data is thorough and of first rate quality, greatly superceding the rather mediocre scRNAseq published earlier for the eye. Moreover, they experimentally validate many of their observations using transgenics and imaging. This is first rate work.

As such, this data will be a rich resource for mathematicians, physicists, and data scientists to explore important questions regarding dynamics of cell state transitions, cell maturation, positional information, and pattern formation. Nature Communications is a terrific forum for this paper given the interdisciplinary bent of the journal.

I think the authors actually do a disservice to themselves and the work by not emphasizing the unique property of the eye disc to capture developmental dynamics in one tissue sample without any inference. This is opposite to scRNAseq of embryos or other organs where either single samples are studied or sparse temporal sampling is performed. These studies rely on heavyhanded and questionable methods to mathematically infer the underlying dynamics. These inferences rarely are validated. Whereas here in this manuscript, the dynamics are beautifully laid out with remarkable temporal resolution. The authors should rewrite some of the introduction, results, and discussion to emphasize the singularity of this work.

We thank the reviewer for the comments on the manuscript. We have made many changes and rewrote much of the introduction, results and discussion to address those concerns.

I also think the authors could easily enhance the impact of their work by some simple linear dimension reduction analysis like PCA. More details below. If they do PCA on subsets of cells as I suggest below, they will definitely be able to publish in this manuscript new insights about gene networks and dynamics of cell maturation and specifiication. If it is fruitful, which I am confident it will be, they could replace some of the more simplistic analysis main figures (Figs 3 and 8) with PCA results. It should only take a couple of weeks to analyze the data by PCA.

We thank the reviewer for this suggestion. We performed PCA on the whole dataset as well as on individual clusters. PCA on the whole dataset did not show a good segregation of cell clusters as observed with UMAP when single cell profiles were projected onto the top 2 principal components of PC space (see figure below). We also performed PCA on individual clusters, including the Convergence cluster. We observe considerable overlap (an average of 71% identity among the top 30 genes for the 12 clusters tested) between the top PC1 genes and the top differentially expressed genes in a given cluster. We include these results in lines 326 to 336 and have also added Supplementary Tables 4 and 5 that show the top 1,000 PC1 genes

with loadings/weights and the eigen values of each PC for a given cluster, respectively.

Below I list a number of points as I came across them in the manuscript. Some are major points and some minor.

INTRODUCTION

The field has traditionally used the acronym R cells for the 8 photoreceptors instead of PR cells. It makes more sense to keep with the tradition in context of discussing R cell specific transcriptomes. etc. Throughout the manuscript.

Good point! We changed PR to R cells throughout the manuscript.

The MF generates a new column every 2 hours not 1.5 hours. This had been inferred in 1980s experiments and validated by Gallagher et al in 2022 (elife) in which they observed it in live discs.

Agreed and thank you for that correction. We changed 1.5 hrs to 2 hrs

It makes more sense to move the sentence about furrow speed to before the one beginning "Posterior to the MF, the R8..." It makes everything following more easy to comprehend

Agreed and we have made the changes as suggested in lines 69-71

There are R3 or R4 specific genes that have been already described in the literature. E(spl)m-delta for example. Correct this error.

We searched the literature and only found one enhancer called ‘mδ0.5’ that shows R4-specific pattern. We could not find that E(spl)m-delta gene expression that is R3 or R4 specific. We have noted in the manuscript that an R4-specific enhancer has been reported.

“The R7 PR and four cone cells are added to each ommatidial cluster during late larval stages “ This sentence is misleading. There is a gradient of differentiation where in late L3 larvae, some ommatidia do not have R7 and cone cells. Perhaps it would be more clear to say the R7 and cone cells are recruited to ommatidia that are located more posterior in the eye disc since these ommatidia are more mature than their anterior neighbors.

Agreed and we have rewritten the text as suggested in lines 71-77.

It states “less mature uncommitted progenitor cells are located anteriorly” This is not strictly correct. The progenitor pool is distributed throughout the eye posterior to the MF - only a minority of progenitors actually differentiate into R cells and cones. There may indeed be stages of progenitor maturation. For example, Yan and Pnt genes are expressed with pulsatile dynamics in progenitors as they age (Pelaez et al 2015 elife and Bernasek et al Biorxiv <https://www.biorxiv.org/content/10.1101/430744v4>). But we cannot really know if this represents maturation since maturation usually is irreversible. This scRNA seq may actually uncover maturation.

Agreed and we have made the changes as suggested in lines 78-79

It is repeatedly stated that there are streams or clusters of cells in the abstract and introduction. They are referring to the dimensionally reduced data representations. Calling them streams or clusters prior to showing the data is misleading. The summary paragraph at the end of the introduction might benefit better from less description of what the low dimensional representations look like and more what conceptual advances are made.

Agreed and we have replaced 'streams' with 'strands' throughout the manuscript. We also made changes to the introduction as per the reviewer’s suggestion.

Figure 1 refers to AUnd and PPN and SMW but the introduction does not mention these developmental processes. Maybe it would be better to cite Figure 1B and B’ in the introduction and also discuss these undescribed processes there. They are described somewhat in the figure legend but it is better for readers to read in main text and the introduction is the appropriate place.

Agreed and we have added AUnd, PPN and SMW to the introduction.

RESULTS

Figure 1 and 2, “streams of cells emerging” implies some sort of dynamics rather than what

UMAP is showing which is a low dimensional shape or geometry of the high dimensional data. Strands is a more appropriate descriptor. Strands connected to clusters is also more accurate. Not emergence. Please replace streams with strands throughout.

Agreed and we have replaced streams with strands throughout the manuscript.

The analysis of R3 and R4 data is fascinating. In addition to finding expected genes like E(spl), they find enrichment of DPR1 in R4 and DIPs in R3. They do not make note of the literature on these proteins - DIPs are Ig-like receptors for DPRs like DPR1. Often found on interacting neuronal processes, here they are expressed in neighboring cells that are known to signal one another. Is this DPR-DIP as signal between the two as well? This bears discussion.

Agreed and we have added discussion about the *Dpr* and *DIP* genes as suggested.

Fig 1E and text uses a standard “pseudotime” algorithm to infer dynamical features from the static data (snapshot collected). Pseudotime inferences are in general very weak and rarely validated. In this present situation, it is completely unnecessary. Since the eye disc has the unique feature of having cells at progressively older stages of development all contained together, there is no need to use pseudotime. Real developmental time is represented in the data. This is really seen by the interconnected strands of datapoints representing the R cells, SMW cells and LM cells. This is exactly what one predicts from developmental data with embedded dynamical features. For point of reference, please see the Tabula Muris data of mouse hematopoietic cells (<https://www.biorxiv.org/content/10.1101/2021.05.03.442465v1>), which is also a system with embedded dynamics. Therefore, the pseudotime analysis should be removed and more effort placed on extracting dynamical features of the data (see PCA below).

We agree that the pseudotime analysis may be less than perfect but we also think that it confirms our interpretation that the R clusters have a temporal component. We moved the pseudotime figure to supplementary as per the reviewer’s suggestion.

The representation of the oldest R cells is a cluster (convergence cluster) in the UMAP embedding. The authors then subject the individual classes of R cell data to selective UMAP embedding. While this is one approach to take, they are probably embedding data that might be much more amenable to linear dimension reduction like PCA. UMAP and its ilk are non-linear DR methods that give us a probabilistic and non-quantitative view of high dimensional data. If each R cells' data cloud is more linear in structure, even if high dimensional, then PCA will not only accurately show the cloud’s structure, but the distances between datapoints, ie. cells, will quantitatively scale with real measures of difference. This is not the case for UMAP or t-SNE. It would be trivial for the authors to run PCA on the selected R cell data (python, R and Matlab all have PCA packages). In fact if they select each R cell type to analyze by PCA, then the first eigenvector (PC1) will uncover the developmental dynamics of that cell type’s maturation program. Genes correlated along that eigenvector having large eigenvalues will be those genes acting in the GRN for maturation of that cell type. The identities and positions of those genes in the GRN will be nicely displayed. The convergence cluster itself could be subject to PCA and genes differentially

expressed in the cluster or biased in expression will be identified. It will settle whether the cluster is a unitary “melded” cell type of mature R cells or is a collection of highly related multimodal cells types (I think it is probably the latter).

PCA of any cluster (ie SMW, MF, PPN, etc) can be done to reveal the hidden dynamics of GRNs working in each class of cells. This is easy to do, and is potentially of great value in such a developmental system since it is physically designed for this kind of analysis.

Great point and we thank the reviewer for this suggestion. Accordingly, we performed PCA on all cell clusters and show the top 1,000 PC1 genes with loading values for each gene in Supplementary Table 4. We also report the eigen values of each PC for each cluster in Supplemental Table 5.

My comments about PCA apply not only to the scRNA data but also scATAC data. This beautiful data shows a similar UMAP embedding representation as scRNA. Classes of cells based on identities could also be subject to PCA as described above for scRNA data. It would then be trivial to correlate genes detected along PC1 of the scRNA data with genes detected along PC1 of the ATAC data. This would help build robust GRNs out of the work. The authors attempt to use SCENIC to start building GRNs but this is a crude tool given the remarkable data at the disposal of the authors.

In this regard, the focus of the manuscript on new “marker gene” identification does disservice to the enormous potential of the eye system to understand the transcriptome dynamics of cell fate transitions and cell state maturation.

As the reviewer has suggested, we performed PCA on all cell clusters of snATAC-seq data. However, we could not identify significant genes correlated between scRNA-seq and snATAC-seq. This may be due to the sparse nature of the snATAC-seq data.

Figure 6B - how were cell identities classified since the data is differential genome reads? Did they look for cells where known marker genes had significantly enriched read counts? More information should be given in the text.

We agree that this should have been made clearer in the text and we have made appropriate changes. In brief, we employed several approaches to classify cell identities in the snATAC-seq data. First, we performed integrative analyses and transferred labels and gene expression values from scRNA-seq to snATAC-seq data. We added details of this approach to the Methods section and added more detail about snATAC-seq annotation. Second, we used previously published cell type-specific eye enhancers to identify clusters (e.g., the *dac* 3EE and 5EE enhancers). Finally, we selected cell type-specific peaks and performed *in vivo* enhancer-reporter analyses to identify and validate the cell type assignments.

DISCUSSION

Line 490: computational representation should be changed to mathematical representation.

Agreed and we have made changes as the reviewer suggested.

Line 509 sentence explains why DV domain cells are not embedded within clusters. Another reason could be that they are embedded but within 3D clusters whereas UMAP embeds in 2D. So a cluster, say dorsal R7s might lie above ventral R7s in 3D space but UMAP squashes both together with its 2D embedding.

Good idea! However, we generated a 3D FeaturePlot for the *mirror* gene and did not see any specific segregation of cells expressing *mirror*. Cells expressing *mirror* are still randomly distributed in the 3D plot.

Lines 553 -568 is a bit naive. Cells are not “losing their identity” as they mature. They are expressing sets of genes necessary for maturation of all photoreceptor cells - axon guidance, rhabdomere morphogenesis, opsins etc. The few highly distinct TFs expressed early in their maturation are being complemented with many more common genes expressed in neurons and photoreceptors, thus obscuring the differences these cells still have with their distinct TFs expressed early and late. The UMAP embedding is a statistical representation, and correlations are stronger if more genes are expressed in common. Hence there is a single cluster. The same naivete can be said about the olfactory and optic lobe papers.

We thank the reviewer for this comment and agree with what the reviewer has suggested. The cells in the convergence cluster may not be losing their identity but it is possible that highly expressing common genes may be obscuring the genes that drive cell type distinction. We added this information to the discussion in lines

Reviewer #2 (Remarks to the Author):

This manuscript describes two datasets: a single-cell mRNA sequencing and a single-cell ATAC sequencing dataset of the late third instar larval eye disc. The authors generated these two datasets, performed standard analysis of the two datasets in R that led to the generation of two UMAP plots, which they then used to assign clusters based on known markers. Then, they use this information to identify cell-type specific markers and cell-type specific open chromatin regions, respectively. Finally, they discuss two observations: a) the convergence of all photoreceptors (PRs) in one state and b) the difference in chromatin accessibility between PRs and cone cells.

The generated datasets appear to be of very good quality, although it would have been necessary for the reviewers to have the data and the scripts available to be able to actually review the paper. However, the paper appears to be just a report of the datasets with minimal, if any, new ideas and knowledge.

Agreed and we have uploaded the raw data, .rds files and the codes.

Moreover, the two observations that are discussed are not well supported and not

convincing.

Regarding the convergence of photoreceptors to a common state:

- We know that the photoreceptors are different with one another, especially R7 and R8, which have different molecular identities, they express different rhodopsins (although this expression comes later, the PRs have to keep their identity and not converge with other PRs). The fact that they converge might be an artifact of the analysis. Instead of re-clustering and remaking the UMAP, the authors should, at least, renormalize all expression data, choose new variable genes (i.e. variable genes only within photoreceptors) and see if they still converge. Even then though, does it make any sense that PRs at L3 lose their individual identity? They haven't made any synapse yet.

We thank the reviewer for this comment. We have performed PCA as suggested by reviewer 1 but did not see isolated and well-defined clusters (see figure above). We also reclustered our data with different UMAP hyperparameters and random seeds and we still observe R cell strands and the Convergence cluster. Supplementary Figure 6 shows cluster plots generated with different hyperparameters. We also subclustered the photoreceptor clusters and still observe convergence. However, we do agree that the identity of each R cell subtype may not be lost but may have been obscured by high levels of expression of axon-related genes. We have added this to the discussion.

- What happens later into pupal stages? How does this convergence evolve especially when the PRs have to make synapses or when the R7 and R8 cells acquire a pale or yellow identity?

We agree that these are good questions and we don't know the biological significance of convergence. It may simply be a reflection of the transcriptional dominance of genes related to axogenesis and synapse formation and have explicitly stated this in the revised manuscript.

- The authors should verify the expression of synaptic genes at the level of the protein as well. Isn't it too early to make synapses? Their postsynaptic targets are not mature to make synapses before P20 or P30.

We agree that at first glance it may be surprising that synaptic genes are being expressed at P0 since synapse formation does not occur until 20 hrs later. However, gene expression at the RNA level needs to occur before the encoded protein is needed and proteins involved in synapse formation need to be expressed before the actual synapse is formed - it does not happen in an instant, or even an hour. It therefore makes sense to us that synaptic genes begin to be expressed at the beginning of pupal development. Moreover, it is possible that these genes are being transcribed but some may not be translated until some hours later. The point we are trying to make is that axogenesis and synapse formation gene expression is a driving force for transcriptional convergence of R cells at late larval/early pupal stages. We have validated expression of several genes in this study and think we have shown very clearly that our scRNA-seq data is very accurate. Respectfully, we

think that verifying the expression of synaptic genes is beyond the scope of this paper and would not add substantially to our major conclusions.

- In any case, what is the “biological importance” of this convergence? What does it mean for the photoreceptor specification? Does it mean that they all have to go through a process that involves axon guidance and neuronal projection? Is this new or unexpected?

We don't know the biological significance of convergence although we do not think it is related to R cell specification - that happens much earlier. It may simply be a reflection of the transcriptional dominance of genes related to axogenesis and synapse formation and have explicitly stated this in the revised manuscript. Since all R cells send out axons during larval development we do think that all R cells must go through this process. Although it may not be unexpected, to our knowledge, transcriptional convergence of R cells has not been reported, so we think this is a new result and have tried to present it more clearly in the revised manuscript.

Regarding the argument about accessibility differences between PRs and cone cells: this is very unclear and needs to be further developed, tested, and justified. What are the potential “functionally important differences”? What function do they serve?

We agree that we do not know if the differences we have observed in the chromatin states of R and cone cells are functionally important. We think the difference is rather striking and merits reporting in this manuscript. However, it is only speculation that the chromatin accessibility differences between PR and cones may play a role in some aspect of their differentiation; we are not stating this as a result. Respectfully, we think that functional testing of this hypothesis is beyond the scope of this paper.

Except for the above, I also have two other points that I find concerning:

- First, the authors rely extensively on the use of UMAP to extract information. This is problematic and can lead to false conclusions and confirmation bias by reading the UMAP selectively (<https://www.biorxiv.org/content/10.1101/2021.08.25.457696v1>). For example, the authors don't mention that R7 cells seem to come from cone cells and not the PUnd cluster, which is, of course, biologically incorrect. They also mention that the UMAP is correlated with the physical eye disc, which is in the eye of the beholder. The extensive use of UMAP plots to draw conclusions can be very misleading and has been done extensively in this manuscript.

We understand the reviewer's concerns about extensive use of UMAP but respectfully reply that we think the data fully justifies our approach. There are dozens of genes with known, published, cell type-specific patterns of expression in this tissue at this stage of development. We have shown many of these as FeaturePlots and DotPlots that strongly validate the cluster assignments we have made and could have shown dozens more. We do not think there can be any reasonable doubt about our cluster assignments. In addition, we state that R7 and cone cells differentiate from posterior undifferentiated cells that exit the second mitotic wave. It is true that R7 and cone cells differentiate from the R7 equivalence group and it is expected that

they appear next to each other. However, we think that the R7 strand in our cluster plot (Figure 1E) may originate from the PUnd cluster since it is in direct contact with PUnd. The fact that it is also in contact with the cone cell cluster is likely a reflection of the close relationship between R7 and cones.

- It is impossible to review the paper without having access to the data, but more importantly to the scripts, especially since most of the bioinformatic analyses are not described in detail.

We agree and have uploaded the raw data, .rds files and the code.

Minor points:

- In Figure 8, the authors argue that PRs cluster separately from other cell types. This is actually not true if we believe the dendrogram. In fact, the Convergence cluster (i.e. a PR cluster) is the outgroup of all clusters. What does this mean?

We respectfully offer that the convergence cluster appears separate from other cell types, perhaps suggesting that the gene regulatory networks in this cluster are different from the other cell types.

- Mentioning the four classes of DE genes in the text without explaining what they actually are is of little use. The authors should either describe them in the text or not mention them at all.

- Mentioning numbers of DE genes without mentioning any cutoff is also of little use. How were the DE genes chosen? Fold-change, p-value, q-value? It is known that Seurat gives overblown numbers of DE genes.

We agree with the reviewer. We removed all classification and provided marker gene tables that can be sorted based on p-value or log2fold change.

- The extensive use of R commands, such as FeaturePlot, CoveragePlot, etc, should be eliminated, as this makes the manuscript inaccessible.

We agree and have made these changes throughout the manuscript.

- The stage of the sequenced eye disk (late L3) should be mentioned in the Abstract

We agree and have made this change.

- In the Abstract, the authors mention that a single-cell genomic atlas will greatly empower the Drosophila eye as a model system. I don't think the eye disk needs such an atlas to be empowered. It has been one of the most commonly used model systems, as the authors also state in the Introduction.

We agree and have rewritten the abstract accordingly.

Reviewer #3 (Remarks to the Author):

In this manuscript the authors present a single-cell multi-omics resource detailing the RNA-Seq and ATAC-Seq profiles of the developing drosophila eye. The noteworthy aspects of this work are 1) the number of cells profiled (~27,000) which suggests coverage of all cell types present in the late larval eye, 2) identification of novel cell type markers, and 3) the interesting developmental pattern whereby photoreceptors emerge from transcriptionally distinct trajectories only to converge upon a similar transcriptional identity. The significance of this work is hinted at in the manuscript because of the richness of the drosophila system for understanding the role of developmental genes and gene regulatory networks such as Pax6, but could have reached a more ambitious potential by comparing wildtype patterns of development to mutants affecting eye development including mutants of the potential regulators identified by SCENIC analyses. The methodology is sound and meets expected standards in the field of single cell transcriptomics and epigenomics. There is sufficient detail for the experiments to be reproduced. This reviewer has no major concerns apart from that this work stands mostly as a resource upon which deeper insights may be gleaned in the future. A minor suggestion for improving the current manuscript would be to condense the part of the manuscript devoted to identifying clusters based on known markers, to leave more room to compare and contrast with the previous single-cell studies performed by this group and others and to emphasize the new found knowledge and understanding of eye development that has been gained by the current study.

We thank the reviewer for the comments on the manuscript. We made a brief comparison with published data and show it as Supplementary Figure 14. We also uploaded our data to GEO and it is accessible for everyone to download and compare with published data.

Reviewer #4 (Remarks to the Author):

In this work, the authors report a single cell genomics atlas of the Drosophila larval eye. Both scRNA-Seq data and scATAC-Seq were generated using the 10X genomics platforms from similar tissues. Overall, I think this will be a useful data resource. The followings are my comments.

1. The variation of biological replicates. The scRNA-Seq dataset was generated from two biological replicates and the scATAC-Seq dataset was generated from three biological replicates. The authors showed through UMAP plots that the major cell types are separated in scRNA-Seq and scATAC-Seq data. In the analysis, the authors focus on the variation of different cell types, which may be confounded by the variation of different biological replicates. It would be great to show UMAP plots, with different colors representing different biological replicates, to see whether the same cell types from different biological replicates are mixed together or not.

We agree and have added plots from both biological replicates to Supplementary Figure 1. The plots show that each cell type is well represented in both replicates.

2. In this work, many conclusions are drawn based on patterns of UMAP plots: e.g. relationship between different cell types. Because UMAP provides a 2D representation of the original dataset and the result will vary depending on the hyperparameter in UMAP and also the random seed. It will be great if the authors can provide the hyperparameters used in UMAP plots. It would also be great if the authors can test different hyperparameters and random seeds in implementing UMAP and check whether the conclusions in the work is rigorous (especially when the random seed in UMAP changes).

We thank the reviewer for this comment. We performed PCA as suggested by reviewer 1 but did not see isolated and well-defined clusters. We also reclustered our data with different UMAP hyperparameters and random seeds and we still observe PR streams and convergence. Supplementary Figure 6 shows cluster plots generated with different hyperparameters. We also subclustered the photoreceptor clusters and still observe convergence. However, we agree that the identity of each PR subtype may not be lost in the Convergence cluster but may have been obscured with high expression of axon-related genes. We have added this information to the discussion.

3. In this work, “In addition to transferring labels, gene expression values from scRNA-seq can also be transferred and imputed on snATAC-seq clusters such that marker gene expression patterns can be visualized on snATAC-seq cluster plots.” This practice may be problematic for validation: both the cell type labels and gene expression are transferred from scRNA-seq, and it is expected to see agreement between the transferred marker gene expression and the transferred cell type. It seems to me that a more reasonable practice is to use gene activity score of marker genes in scATAC-Seq to validate the cell type labels transferred from scRNA-Seq data.

We thank the reviewer for these comments. We have used gene activity scores of marker genes in snATAC-seq to validate the cell type labels and have made appropriate changes to the Methods section and added more detail about snATAC-seq annotation. Furthermore, we employed more than one approach to classify the cell identities of the snATAC-seq data. First, we performed integrative analyses and transferred labels and gene expression values from scRNA-seq to snATAC-seq data. Second, we used previously published cell type-specific eye enhancers to identify the clusters (e.g., the *dac* 3EE and 5EE enhancers). Finally, we selected cell type-specific peaks and performed *in vivo* enhancer-reporter analyses to identify and validate the cell type assignments.

4. The raw and processed datasets should be accessible to the reviewers. Reviewer tokens in GEO should be given for reviewers to check that the datasets are publicly available.

We have uploaded the .rds files and the code.

REVIEWER COMMENTS

Reviewer #1 (Remarks to the Author):

The authors have done an excellent job addressing all of my concerns. It should be accepted for publication.

Reviewer #2 (Remarks to the Author):

Please find my review as an attachment, as I would like to add some Figures, which I cannot add in this space.

Reviewer #4 (Remarks to the Author):

All my comments have been addressed reasonably. I do not have any further comments.

After re-reading the manuscript and exploring the data, my concerns unfortunately stand.

- 1) I am not convinced that the Convergence cluster is real. The cluster has lower number of Features per cell and Counts per cell than any other cluster (except for the AUnd in terms of Counts)

This can explain why the authors do not see many of the cell type-specific genes. I agree that the cells are flooded with mRNAs relating to axogenesis and synapse formation, which in combination with the normalization and the lower recovery of genes from these cells does not allow them to see cell-type specific genes. In fact, many other genes can't be found in the Convergence cluster, such as *Elav*:

The authors should perform an *in situ* hybridization against *elav* to show that indeed it is not expressed in mature photoreceptors. Of course, the protein is there (Figure 3F) and it also remains later, so it would be very surprising to not see it.

- 2) The use of UMAP to make conclusions remains unnecessarily high. I never doubted the cluster assignment, which indeed relies on markers, but I am not convinced by the interpretations that come from the UMAP itself. As I mentioned in my first review, an obvious interpretation that comes from the UMAP is that R7 cells come from the cone cells and not from the PUnd. In fact, the R7 cells are closer to the cones in the UMAP

than R7 and R8 cells are to the Convergence cluster. Also, R1 and R6 also come from the R7 equivalence group, but they are very distinct in the UMAP, which is another reason why UMAP is unreliable to make conclusions without testing *in vivo*; the interpretations are completely subjective.

For the above reason, I do not believe that “the data sets can be used as a virtual *in situ*”. On top of that, as the authors mention, the datasets lack DV axis information, which *in situ* hybridization offers. Also, their dataset cannot resolve mature photoreceptors from one another, whereas this is possible by FISH. Finally, the incomplete coverage of single-cell sequencing data, in combination with normalization processes, may lead to distorted views, such as the fact that Elav is not expressed in the mature photoreceptors.

There have been other ways published that could generate “virtual *in situ*” (e.g. Novosparc), but UMAP is definitely not one.

- 3) I, finally, am also not convinced about the significance of the chromatin accessibility differences between R cells and non-neuronal cells, but this is minor compared to concerns 1 and 2. Nevertheless, since the authors cannot conclude anything from the paragraph and can only speculate, I am not sure I see the reason for it. On that note, Figure 7 is named “Chromatin is more accessible in photoreceptors compared to cones.”, but it doesn’t show this: in fact, it shows two loci, one is more accessible in neurons and one is more accessible in cones.

Reviewer 2 Comments

After re-reading the manuscript and exploring the data, my concerns unfortunately stand.

1) I am not convinced that the Convergence cluster is real. The cluster has lower number of Features per cell and Counts per cell than any other cluster (except for the AUnd in terms of Counts)

This can explain why the authors do not see many of the cell type-specific genes. I agree that the cells are flooded with mRNAs relating to axogenesis and synapse formation, which in combination with the normalization and the lower recovery of genes from these cells does not allow them to see cell-type specific genes. In fact, many other genes can't be found in the Convergence cluster, such as *Elav*:

The authors should perform an *in situ* hybridization against *elav* to show that indeed it is not expressed in mature photoreceptors. Of course, the protein is there (Figure 3F) and it also remains later, so it would be very surprising to not see it.

We thank the reviewer and agree with the reviewer's comments. We agree that the Convergence cluster cells are not novel cell types or represent a new identity; instead, we agree that the high expression of axogenesis and synapsis genes is obscuring the expression of cell type-specific genes. Therefore, as pointed out by the reviewer, we agree that the Convergence cluster may be an artifact of UMAP. We added this information to the manuscript.

2) The use of UMAP to make conclusions remains unnecessarily high. I never doubted the cluster assignment, which indeed relies on markers, but I am not convinced by the interpretations that come from the UMAP itself. As I mentioned in my first review, an obvious interpretation that comes from the UMAP is that R7 cells come from the cone cells and not from the PUnd. In fact, the R7 cells are closer to the cones in the UMAP than R7 and R8 cells are to the Convergence cluster. Also, R1 and R6 also come from the R7 equivalence group, but they are very distinct in the UMAP, which is another reason why UMAP is unreliable to make conclusions without testing *in vivo*; the interpretations are completely subjective.

For the above reason, I do not believe that "the data sets can be used as a virtual *in situ*". On top of that, as the authors mention, the datasets lack DV axis information, which *in situ* hybridization offers. Also, their dataset cannot resolve mature photoreceptors from one another, whereas this is possible by FISH. Finally, the incomplete coverage of single-cell sequencing data, in combination with normalization processes, may lead to distorted views, such as the fact that *Elav* is not expressed in the mature photoreceptors.

There have been other ways published that could generate "virtual *in situ*" (e.g. Novosparc), but UMAP is definitely not one.

We thank the reviewer and agree with his comments. We agree that reliable conclusions concerning spatial gene expression cannot be derived from UMAP. We made appropriate changes to the manuscript and removed all sentences that include "virtual *in situ*"

3) I, finally, am also not convinced about the significance of the chromatin accessibility differences between R cells and non-neuronal cells, but this is minor compared to concerns 1 and 2. Nevertheless, since the authors cannot conclude anything from the paragraph and

can only speculate, I am not sure I see the reason for it. On that note, Figure 7 is named "Chromatin is more accessible in photoreceptors compared to cones.", but it doesn't show this: in fact, it shows two loci, one is more accessible in neurons and one is more accessible in cones.

We thank the reviewer for the comment and agree that there is not an obvious difference between PRs and cones - each has specific ATAC-seq peaks. Accordingly, we edited the manuscript so that we only report photoreceptor- or cone-specific peaks and removed statements suggesting chromatin is more accessible in cones than photoreceptors.

Overall, comments from Reviewer 2 were insightful, helpful, and have improved this manuscript. Thank you.

REVIEWERS' COMMENTS

Reviewer #2 (Remarks to the Author):

The authors have addressed my concerns.

However, this has led to some inconsistencies in the manuscript, e.g.

line 573: "it seems unlikely that the Convergence cluster is an artifact of dimension reduction."

line 591: "non-linear dimension reduction using UMAP may be introducing Convergence cluster as a computational artifact."

Moreover, the Abstract is not corrected to include their revisions, e.g.

line 29: "we observe that cone cells show more cell type-specific peaks than photoreceptors."

I would therefore recommend to proof-read the text to eliminate such inconsistencies.

As a final note, I would personally edit the text more to address the concern about the Convergence cluster, especially since the authors think that "the Convergence cluster may be an artifact of UMAP" and a lot of their results and discussion center around this potentially artifactual cluster. However, this is a personal opinion and it's the authors' choice to decide.

Reviewer #2 comments

The authors have addressed my concerns.

However, this has led to some inconsistencies in the manuscript, e.g.

line 573: "it seems unlikely that the Convergence cluster is an artifact of dimension reduction."

line 591: "non-linear dimension reduction using UMAP may be introducing Convergence cluster as a computational artifact."

We have removed the text and made changes to remove all inconsistencies from the manuscript.

Moreover, the Abstract is not corrected to include their revisions, e.g.

line 29: "we observe that cone cells show more cell type-specific peaks than photoreceptors."

We have changed the sentence to "we observe that the chromatin accessibility between cones and photoreceptors is distinct". This distinction between cones and photoreceptors is an observation in our snATAC-seq data and we are reporting it in the manuscript. We are not implying any interpretation on this observation.

I would therefore recommend to proof-read the text to eliminate such inconsistencies.

We have made changes to remove all inconsistencies from the manuscript.

As a final note, I would personally edit the text more to address the concern about the Convergence cluster, especially since the authors think that "the Convergence cluster may be an artifact of UMAP" and a lot of their results and discussion center around this potentially artifactual cluster. However, this is a personal opinion and it's the authors' choice to decide.

We have modified the interpretation and conclusion about this cluster in all sections of the manuscript and hope that our edits satisfactorily address all concerns.